# Lie-Algebraic Acceleration of Neural Koopman Dynamics

**Jongwon Lee** [1]   **Jiwoong Kim** [1]   **Jungwoo Park** [1]   **Sungwoo Park** [1] [†]

## Abstract

We present a Lie-algebraic approach to model Koopman dynamics that integrates algebraic structure with computational scalability. The proposed formulation constrains the neural generators to evolve within prescribed Lie subalgebras and constructs finite-time flows through a neural Magnus expansion construction, thereby maintaining consistency with the associated Lie-group composition over each time segment. To address the computational burden inherent in sequential propagation, we exploit the associativity of Lie-group compositions and construct segmentwise propagators via a prefix-scan algorithm, which reduces the depth of temporal composition from linear to logarithmic. Consequently, the framework enables accurate long-horizon prediction while improving computational efficiency, and provides a principled foundation for scalable Koopman operator learning for nonlinear systems.

## 1. Introduction

The Koopman operator provides a linear representation of nonlinear dynamical systems by acting on observables, and provides a powerful framework for analyzing and predicting complex dynamical behaviors (Koopman, 1931; Mezić, 2005). Its utility has been demonstrated across diverse domains where direct state-space modeling is difficult or where physically meaningful observables are accessible, including climate variability (Valva & Gerber, 2025), molecular dynamics (Nüske & Klus, 2023), plasma physics (Ghannadian et al., 2025), and materials discovery (Peng et al., 2022).

In the realm of artificial intelligence and machine learning, data-driven Koopman methods have expanded this classical framework through latent-space representations and neural operator architectures (Xiong et al., 2024; Meng et al., 2024; Naiman et al., 2024; Wang et al., 2023; Liu et al., 2023; Singh et al., 2024; Zhang et al., 2025). These approaches have substantially improved expressivity and modeling capacity. Despite these advances, their black-box parameterizations often obscure interpretability and suffer from representational limitations when capturing nonlinear dynamics. Moreover, existing formulations rely on the sequential composition of short-time propagators, which introduces computational inefficiency along the temporal axis and instability in long-horizon prediction.

These representational and stability issues in existing Koopman formulations motivate an approach grounded in algebraic structure that offers a principled alternative that reflects the geometric and physical constraints of dynamical systems. Embedding algebraic structure directly into the model constrains admissible trajectories, enforces conservation laws, and promotes stable numerical integration. This perspective has long been essential in domains such as fluid mechanics (Arnold, 1966), molecular dynamics (Tuckerman et al., 1992), celestial mechanics (Wisdom & Holman, 1991), and plasma physics (Morrison & Greene, 1980), where alignment with intrinsic symmetries is the key to both theoretical fidelity and computational robustness.

In this spirit, we develop a time-varying Koopman framework grounded in Lie groups and Lie algebras that yields improved efficiency along both spatial and temporal axes. We cast the generator–propagator relationship through a neural Magnus expansion (Blanes et al., 2009) so that the learned finite-time flow remains within the appropriate Lie group. To retain structural fidelity while keeping computation tractable, we restrict the infinitesimal generator to a Lie subalgebra of the general linear algebra, exploiting closure and symmetry. However, this Lie-theoretic formulation alone fails to resolve the inherent sequential bottleneck arising when composing multiple finite-time propagators. To address this computational challenge, we necessarily leverage the associativity and transitivity properties of Lie-algebraic composition, enabling parallelization of operator products via a *prefix-scan* (Blelloch, 1989). To the best of our knowledge, these elements yield the first Koopman learning framework that integrates Lie-theoretic structure with temporal prefix-scan parallelization.

In summary, the main contributions of this work are twofold:

1. **Lie-algebraic Koopman Dynamics.** We propose a time-

[†]Corresponding author.   [1]Department of Computer Science and Engineering, Korea University. Correspondence to: Sungwoo Park <sungwoo_park@korea.ac.kr>.

*Proceedings of the 43$^{rd}$ International Conference on Machine Learning*, Seoul, South Korea. PMLR 306, 2026. Copyright 2026 by the author(s).

varying Koopman operator framework grounded in Lie group and Lie algebra theory, using a neural Magnus expansion to construct finite-time propagators that preserve algebraic composition. The generator is constrained to a Lie subalgebra of the general linear algebra, which reduces computational cost and numerical error while improving interpretability through algebraic symmetry.

2. **Acceleration of Scientific Simulation.** To further enhance scalability, we introduce a prefix-scan–based temporal parallelization scheme that achieves logarithmic-depth composition and enables efficient long-horizon propagation. Together, these elements establish a coherent algebraic framework for stable and physically meaningful learning of dynamical systems.

**Problem Statement.** The task of approximating the long-term evolution of complex dynamical systems from prescribed initial conditions constitutes a fundamental problem in scientific computing. We formulate this task within the general framework of an initial value problem. Let the system state be denoted by $\mathbf{x}(t) \in \mathcal{X} \subseteq \mathbb{R}^d$ and assume its evolution is governed by a time-dependent vector field $\mathrm{v}_t : \mathcal{X} \to \mathbb{R}^d$:

$$\dot{\mathbf{x}}(t) = \mathrm{v}_t\big(\mathbf{x}(t)\big), \quad \mathbf{x}(0) = \mathbf{x}_0, \quad t \in [0, T]. \quad (1)$$

Given an initial state $\mathbf{x}_0$, the objective is to approximate the time-evolving system state $\mathbf{x}(t) = \mathbf{F}^{t,0}(\mathbf{x}_0)$, where $\mathbf{F}^{t,0} : \mathcal{X} \to \mathcal{X}$ denotes the flow map that maps the initial condition to the system state at time $t$. However, direct approximation may be challenging when the vector field, *i.e.*, $\mathrm{v}_t$, and the induced flow exhibit strong nonlinearity or are not fully characterized. This motivates extending our perspective beyond a purely state-space approach to leverage the underlying geometric and algebraic structures governing temporal evolution.

## 2. Lie-algebraic Koopman Dynamics

We now present a unified algebraic framework for learning Koopman operators, which leverages Lie-theoretic formulations to achieve scalable computation.

**Operator-theoretic Koopman Dynamics.** In this section, our first aim is to algebraically reformulate the Koopman dynamics. Let us denote $\mathbf{F}^{t,s} : \mathcal{X} \to \mathcal{X}$ as the flow map advancing the state from time $s$ to $t$, $\mathbf{x}(t) = \mathbf{F}^{t,s}(\mathbf{x}(s))$. The Koopman formalism provides a linear, albeit infinite-dimensional, perspective on this nonlinear system. Given any observable $\varphi \in \mathcal{G}(\mathcal{X})$, the *time-varying Koopman operator family* $\mathcal{K}^{t,s}$ acts by

$$\big[\mathcal{K}^{t,s}\varphi\big](\mathbf{x}) = \varphi\big(\mathbf{F}^{t,s}(\mathbf{x})\big), \quad (2)$$

and satisfies the transitivity property $\mathcal{K}^{t,u}\mathcal{K}^{u,s} = \mathcal{K}^{t,s}$ for $0 \leq s \leq u \leq t \leq T$. The infinitesimal generator $\mathcal{L}_t$

associated with the Koopman operator can be defined via Lie derivative along the vector field $\mathrm{v}_t$:

$$\frac{\mathrm{d}}{\mathrm{d}t}\big[\mathcal{K}^{t,s}\varphi\big](\mathbf{x}) = \big(\mathcal{L}_t\varphi\big)\big(\mathbf{F}^{t,s}(\mathbf{x})\big), \quad \mathcal{L}_t\varphi = \mathrm{v}_t \cdot \nabla\varphi. \quad (3)$$

This relationship can equivalently be expressed in operator form as $\partial_t\mathcal{K}^{t,s} = \mathcal{L}_t\mathcal{K}^{t,s}$, whose formal solution involves the time–ordered exponential (Baake & Schlaegel, 2011; Iserles et al., 2000; Dyson, 1949):

$$\mathcal{K}^{t,s} = \mathcal{T}\exp\left(\int_s^t \mathcal{L}_\tau \mathrm{d}\tau\right), \quad (4)$$

where $\mathcal{T}$ indicates the time-ordering operator required to correctly account for the noncommutativity. Conceptually, the exponential map bridges local and global dynamics by mapping infinitesimal *generators* $\{\mathcal{L}_\tau\}_{\tau \in [s,t]}$ in the space of derivations into finite-time *propagators* $\{\mathcal{K}^{t,s}\}$ within the corresponding space of automorphisms. Short-time steps $\exp(\Delta t\mathcal{L}_{t_k})$ assemble into a valid global evolution through this integration procedure. Guided by this generator–propagator correspondence, we adopt a generator–centric modeling principle: instead of directly parameterizing the global Koopman operator, we model the evolution locally through the infinitesimal generators $\mathcal{L}_t$ (Pazy, 1983; Mezić, 2013; Mauroy et al., 2020).

**Algebraization of Koopman Dynamics.** To algebraically ground the generator–propagator perspective, we briefly recall the fundamental *Lie group–Lie algebra* correspondence. A Lie group $G$ is a smooth manifold endowed with a continuous group operation, *i.e.*, $\circ$, whose elements represent transformations or symmetries of a given system (Hall, 2015). Its associated Lie algebra $\mathfrak{g}$ corresponds to the infinitesimal counterpart, identified with the tangent space at the identity element, and is equipped with a Lie bracket $[\cdot, \cdot] : \mathfrak{g} \times \mathfrak{g} \to \mathfrak{g}$ that measures the non-commutativity of infinitesimal transformations. Then, the exponential map serves as a bridge between these two structures, mapping infinitesimal generators in $\mathfrak{g}$ into transformations in $G$, *i.e.*, $\exp : \mathfrak{g} \to G$ where $X \mapsto e^X := \exp(X)$.

In our naive modeling of Koopman dynamics, the Lie group corresponds to the space of finite-time propagators $\{\mathcal{K}^{t,s}\} \subseteq G \subseteq GL(K)$, and the Lie algebra corresponds to the space of infinitesimal generators $\{\mathcal{L}_t\} \subseteq \mathfrak{g} \subseteq \mathfrak{gl}(K)$. Specifically, we parameterize the local evolution through Lie algebra elements $\mathcal{L}_t$ and recover global propagators via time-ordered exponentiation. To rigorously formalize the algebraic structures underlying the infinitesimal generators and clearly establish the necessary mathematical framework, we introduce the following precise definitions. Let $\{\mathcal{L}_t\}_{t \in I} \subseteq \mathrm{End}(\mathbb{R}^d)$ be a family of infinitesimal generators on a time interval $I \subseteq \mathbb{R}$. A finite-dimensional subalgebra $B = \mathrm{span}\{g_1, \ldots, g_K\}$ is called an *invariant dictionary* if $\mathcal{L}_t(B) \subseteq B$. Equivalently, $B$ is a $\mathfrak{g}$-*module algebra* for

$\mathfrak{g} = \mathrm{span}\{\mathcal{L}_t : t \in I\}$: for any $f, g \in B$ and $t \in I$, the Leibniz rule holds, *i.e.*, $\mathcal{L}_t(fg) = (\mathcal{L}_t f)g + f(\mathcal{L}_t g)$.

Given the basic definitions described above, our first aim is to effectively parameterize the infinitesimal generator. For this purpose, we adopt the *Magnus expansion* (Blanes et al., 2009), an algebraic tool that expresses the exact solution of linear operator differential equations directly in terms of the Lie algebra structure. Specifically, we introduce a neural parameterization of the Magnus expansion, leading naturally to the definition of the Lie-algebraic Neural ODE:

---

**Definition 2.1** (Neural Magnus ODE). Let $\mathfrak{g}$ be a chosen Lie subalgebra with basis $\{\mathfrak{g}_a\}_{a=1}^{|\mathfrak{g}|}$, and $\boldsymbol{\alpha}_{(\cdot)} : [0, T] \times \mathcal{X} \times \Theta \to \mathcal{X}$ be a smooth function parametrized by $\theta \in \Theta$ called a *neural coefficient*. Then, we define an infinitesimal generator as follows:

$$\mathbb{A}_\theta(t) = \sum_{a=1}^{|\mathfrak{g}|} \alpha_a(t, \mathbf{x}_0; \theta)\mathfrak{g}_a = \sum_{a=1}^{|\mathfrak{g}|} \boldsymbol{\alpha}_\theta \mathfrak{g}_a \in \mathfrak{g}.$$

Given the Bernoulli numbers $\mathrm{B}_m$, the algebraic dynamics consists of the system of differential equations:

$$\dot{\boldsymbol{\Omega}}_\theta(t, s) = \sum_{m=0}^{\infty} \frac{\mathrm{B}_m}{m!} \mathrm{Ad}_{\Omega_\theta(t,s)}^m \big(\mathbb{A}_\theta(t)\big), \qquad (5)$$

$$\mathbf{x}(t) = \mathcal{K}_\theta^{t,s} \mathbf{x}(s) = \mathrm{expm}\big(\boldsymbol{\Omega}_\theta(t, s)\big)\mathbf{x}(s). \qquad (6)$$

---

As can be seen, the Neural Magnus ODE provides a rigorous and systematic solution via the Magnus expansion, explicitly connecting the parameterized infinitesimal generator $\mathbb{A}_\theta(t)$ with the global Koopman operator $\mathcal{K}_\theta^{t,s}$. Through this algebraic formulation, the Koopman operator acts as a propagator that naturally advances particle states in accordance with the underlying Lie-algebraic geometry, ensuring accurate and efficient evolution across the defined interval.

More explicitly, this finite-time propagator from fixed time $t$ to $t + \Delta t$ can be expressed as the exponential of a Lie-algebraic element: $\boldsymbol{\Omega}_\theta(t + \Delta t, t) = \sum_{m=1}^{\infty} \boldsymbol{\Omega}_\theta^{(m)}(t, \Delta t)$. Each term $\boldsymbol{\Omega}_\theta^{(m)}(t, \Delta t)$ captures the $m$-th order contribution of the Magnus expansion, encoding nested commutator effects of the infinitesimal generator $\mathbb{A}_\theta(\tau)$ over the time window $[t, t + \Delta t]$. For instance, explicit expressions up to third order are given by:

$$\boldsymbol{\Omega}_\theta^{(1)} = \int_t^{t+\Delta t} \mathbb{A}_\theta(\tau)d\tau,$$

$$\boldsymbol{\Omega}_\theta^{(2)} = \frac{1}{2} \int_{\Delta_{(2)}} \mathrm{Ad}_{\mathbb{A}_\theta(\tau_1)}\big(\mathbb{A}_\theta(\tau_2)\big)d\tau_2 d\tau_1,$$

$$\boldsymbol{\Omega}_\theta^{(3)} = \frac{1}{6} \int_{\Delta_{(3)}} \Big[\mathrm{Ad}_{\mathbb{A}_\theta(\tau_1)}\mathrm{Ad}_{\mathbb{A}_\theta(\tau_2)}\big(\mathbb{A}_\theta(\tau_3)\big)$$

$$+ \mathrm{Ad}_{\mathbb{A}_\theta(\tau_3)}\mathrm{Ad}_{\mathbb{A}_\theta(\tau_2)}\big(\mathbb{A}_\theta(\tau_1)\big)\Big]d\tau_3 d\tau_2 d\tau_1,$$

where the integration domains $\Delta_{(2)} = \{(\tau_1, \tau_2) \mid t \le \tau_2 \le \tau_1 \le t + \Delta t\}$ and $\Delta_{(3)} = \{(\tau_1, \tau_2, \tau_3) \mid t \le \tau_3 \le \tau_2 \le \tau_1 \le t + \Delta t\}$ represent the standard time-ordered simplices.

**Algebraic Structure of Lie Subalgebras.** The Magnus expansion inherently involves intricate time-ordered integrals and nested commutators, rendering direct computation analytically complex and numerically challenging. To simplify this complexity, we impose an additional algebraic structure by selecting an appropriate finite-dimensional Lie subalgebra $\mathfrak{h} \subseteq \mathfrak{g} \coloneqq \mathfrak{gl}(K)$. Then, a basis $\{\mathfrak{h}_a\}_{a=1}^{|\mathfrak{h}|}$ of $\mathfrak{h}$ is called *Lie-closed* if the Lie bracket of basis elements can be expressed as a linear combination of basis elements:

$$[\mathfrak{h}_a, \mathfrak{h}_b] = \sum_{c=1}^{|\mathfrak{h}|} \mathbb{C}_{ab}^c \mathfrak{h}_c, \quad 1 \le a, b \le |\mathfrak{h}| \coloneqq \mathrm{Dim}(\mathfrak{h}).$$

The scalars $\mathbb{C}_{ab}^c \in \mathbb{R}$ are uniquely determined and are called the *structure constants* of $\mathfrak{h}$, which satisfy the following fundamental algebraic constraints for all indices $1 \le a, b, c, e \le |\mathfrak{h}|$:

$$\mathbb{C}_{ab}^c = -\mathbb{C}_{ba}^c \quad \textbf{(Antisymmetry)},$$

$$\mathbb{C}_{ab}^d \mathbb{C}_{dc}^e + \mathbb{C}_{bc}^d \mathbb{C}_{da}^e + \mathbb{C}_{ca}^d \mathbb{C}_{db}^e = 0 \quad \textbf{(Jacobi identity)}.$$

These constraints ensure the internal consistency and algebraic integrity of the Lie algebra structure formed by the basis elements. Consequently, any algebraic parameterization of the infinitesimal generator as $\mathbb{A}_\theta(t) = \sum_{a=1}^{|\mathfrak{h}|} \boldsymbol{\alpha}_\theta \mathfrak{h}_a$ inherits and faithfully preserves the Lie algebra structure encoded by the structure constants. To be more specific, our neural parameterization also inherits this algebraic structure explicitly, leading to restructured tractable representation of the neural Magnus ODE in Definition 2.1, as stated in the following proposition:

---

**Proposition 2.2.** *For the solution to Magnus ODE in* (5), *i.e.*, $\Omega(t, s) = \sum_{a=1}^{|\mathfrak{h}|} \omega^a(t, s)\mathfrak{h}_a$, *associated with the basis elements* $\{\mathfrak{h}_a\} \subset \mathfrak{h}$, *one has*

$$\dot{\omega}^c(t, s) = \alpha_c(t, \mathbf{x}_0; \theta) + \sum_{m=1}^{\infty} \frac{\mathrm{B}_m}{m!} \cdot \Bigg[ \sum_{a_1, \dots, a_m}^{|\mathfrak{h}|} \sum_{b=1}^{|\mathfrak{h}|}$$

$$\Big(\prod_{k=1}^{m} \omega^{a_k}(t, s)\Big) \cdot \alpha_b(t, \mathbf{x}_0; \theta)\mathbb{C}_{a_1 b}^{d_1}\mathbb{C}_{a_2 d_1}^{d_2} \cdots \mathbb{C}_{a_m d_{m-1}}^c \Bigg].$$

---

The algebraic parameterization in Proposition 2.2 explicitly encodes Lie algebra constraints into the neural Magnus ODE, ensuring structured dynamics. To systematically simplify the Koopman operator (5) and (6) by harnessing the structure constants in Lie subalgebra, we employ the Zassenhaus factorization (Kimura, 2017; Casas et al., 2012) to lead the *Lie-algebraic Koopman operator*:

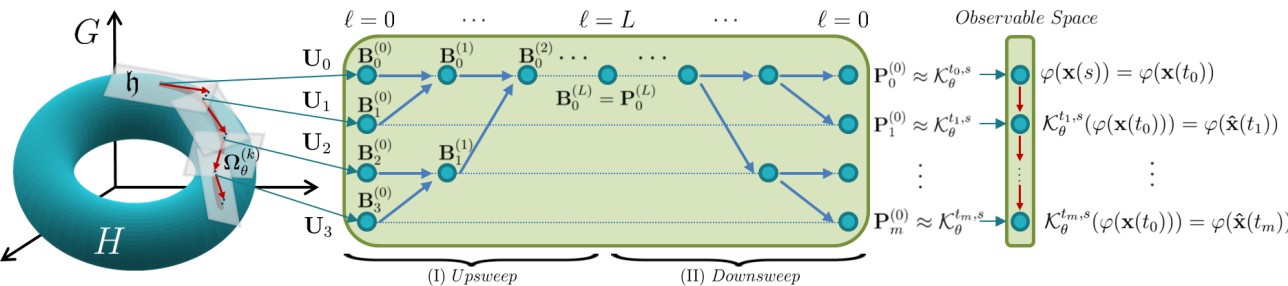

*Figure 1.* **Conceptual Illustration of Parallel Scan on Lie Algebra.** Each red arrow corresponds to $\boldsymbol{\Omega}_\theta^{(k)}$ in (8), matches each evolution $\mathbf{U}_m$ in (9). Second and third figure depicts the overall structure of the Koopman associative scan and observable propagation in our models. Each propagator is compressed into a single complete cumulative product $\mathbf{B}_0^{(L)}$ in the upsweep step, and distributed in the downsweep step, providing each cumulative propagators $\{\mathbf{P}_m^{(0)}\}$, which corresponds to $\mathcal{K}_\theta^{t_m,s}$ for the evolution in observable spaces.

$$\mathcal{K}_\theta^{t,s} = \Big( \prod_{a=1}^{|\mathfrak{h}|} \exp\left(\omega^a \mathfrak{h}_a\right) \Big) \cdot \exp\Big( -\frac{1}{2} \sum_{a<b} \omega^a \omega^b \sum_c \mathbb{C}_{ab}^c \mathfrak{h}_c \Big)$$

$$\cdot \exp\Big( \frac{1}{6} \sum_{a,b,c} \omega^a \omega^b \omega^c \sum_{d,e} \left(\mathbb{C}_{bc}^d \mathbb{C}_{ad}^e + \mathbb{C}_{ca}^d \mathbb{C}_{bd}^e\right) \mathfrak{h}_e \Big) + \mathcal{O}(\|\omega\|^4).$$

Given the newly suggested Koopman operator described above, our objective is to learn the best parameterization $\theta^*$ that captures accurate Koopman dynamics and solves initial value problems:

$$\mathcal{J}(\theta) := \mathbb{E}_{\hat{\mathbf{x}} \sim \mathbb{P}_{\text{data}}} \left[ \int_{[0,T]} \|\mathcal{K}_\theta^{t,0} \hat{\mathbf{x}}(0) - \hat{\mathbf{x}}(t)\|^2 dt \right]. \quad (7)$$

Despite its structured form, directly estimating $\mathcal{K}_\theta^{t,0}$ for all $t \in [0, T]$ generally incurs substantial computational costs. To remedy this, the remainder of this paper addresses precisely this computational challenge, providing a structured algebraic framework that enables numerical efficiency.

## 3. Parallel Scan on Lie Algebra

This section introduces an efficient parallel composition scheme. Consider a partition of the interval $[s, t]$ given by $s = t_0 < t_1 < \cdots < t_M = t$ with corresponding step sizes defined as $h_k := t_{k+1} - t_k$. Leveraging the previously introduced Neural Magnus ODE formulation, we first approximate the Magnus operator over each subinterval $[t_k, t_{k+1}]$ using a spatio-temporal Euler discretization:

$$\boldsymbol{\Omega}_\theta^{(m)} := \boldsymbol{\Omega}_\theta(t_{m+1}, t_m) \in \mathfrak{h} \subseteq \mathfrak{g}, \quad (8)$$

$$\mathbf{U}_m := \exp\left(\boldsymbol{\Omega}_\theta^{(m)}\right) = \mathcal{K}_\theta^{t_{m+1}, t_m} \in H \subseteq G, \quad (9)$$

This formulation naturally leads to an approximate discrete evolution represented by the ordered product of these segment-level propagators:

$$\mathcal{K}_\theta^{t,s} \approx \mathbf{U}_{M-1} \circ \mathbf{U}_{M-2} \circ \cdots \circ \mathbf{U}_1 \circ \mathbf{U}_0, \quad \mathbf{U}_0 = \mathbf{I}_H,$$

where $\circ$ denotes the Lie group operation as introduced in our Koopman operator framework. Because the propagators form a Lie group under group operation $\circ$, it automatically inherit two essential algebraic properties: *transitivity*

$\mathcal{K}_\theta^{t,u} \circ \mathcal{K}_\theta^{u,s} = \mathcal{K}_\theta^{t,s}$ and *associativity* $(\mathbf{U}_c \circ \mathbf{U}_b) \circ \mathbf{U}_a = \mathbf{U}_c \circ (\mathbf{U}_b \circ \mathbf{U}_a)$. These properties provide the exact algebraic foundation required for efficient parallel composition schemes such as **prefix parallel scan** (Blelloch, 1989). In particular, associativity guarantees that the discrete propagator chain can be re-parenthesized without altering its outcome, enabling scalable prefix composition via balanced-tree scans. Building on this, we next describe a three-stage procedure (*i.e.*, *upsweep*, *downsweep*, and *prefix*) for computing the propagators in a temporally parallel manner.

**Parallel Scan on Lie Algebra.** Given the sequence of segment-level propagators in (9), the goal of the parallel scan is to efficiently compute all right-to-left cumulative products prefixes defined as:

$$\Pi(m) := \mathbf{U}_{m-1} \circ \mathbf{U}_{m-2} \circ \cdots \circ \mathbf{U}_0, \quad 1 \le m \le M.$$

To facilitate a balanced binary-tree structure required for efficient parallel computation, we first pad the sequence of operators to a length $M' := 2^{\lceil \log_2 M \rceil}$ by introducing identity elements, i.e., $\mathbf{U}_k := \text{Id}_G$ for $M < k < M'$. Then, we denote the number of hierarchical levels in our binary tree as $L := \log_2 M'$ and proceed to compute the cumulative propagators through the following structured three-step procedure, beginning with the upsweep phase:

---

*(I) Upsweep (Reduction step).* Given the padded sequence of operators $\{\mathbf{U}_j\}_{j=0}^{M'-1}$, we construct a balanced binary tree by initializing its leaves as $\mathbf{B}_j^{(0)} := \mathbf{U}_j$ for $0 \le j < M'$. Moving up the tree, we iteratively compute pairwise products at each higher level. Specifically, for levels $\ell = 0, 1, \ldots, L-1$, each internal node aggregates the results of its two immediate children in the following recursion:

$$\mathbf{B}_j^{(\ell+1)} := \mathbf{B}_{2j+1}^{(\ell)} \circ \mathbf{B}_{2j}^{(\ell)}, \quad 0 \le j < M'/2^{\ell+1}.$$

By construction, each node $\mathbf{B}_j^{(\ell)}$ compactly encodes the product of a contiguous subsequence of propagators:

$$\mathbf{B}_j^{(\ell)} = \mathbf{U}_{(j+1)\cdot 2^\ell - 1} \circ \mathbf{U}_{(j+1)\cdot 2^\ell - 2} \circ \cdots \circ \mathbf{U}_{j \cdot 2^\ell}$$

Thus, at the root node of the tree ($\ell = L$), we efficiently obtain the complete cumulative product:

$$\mathbf{B}_0^{(L)} = \mathbf{U}_{M'-1} \circ \mathbf{U}_{M'-2} \circ \cdots \circ \mathbf{U}_0.$$

This reduction step compresses the propagator sequence into a hierarchical form, setting the stage for efficient parallel distribution in the subsequent downsweep step.

*(II) Downsweep (Distribution step).* In this phase, we efficiently distribute the cumulative product computed at the root node back down through the binary tree to simultaneously recover all individual prefix products. We initiate this step by assigning the group identity to the root of the downsweep structure, $\mathbf{P}_0^{(L)} := \mathrm{Id}_H$. Then, propagating downwards through levels $\ell = L - 1, \ldots, 0$, we iteratively distribute the computed products to each node at level $\ell$:

$$\mathbf{P}_{2j}^{(\ell)} := \mathbf{P}_j^{(\ell+1)}, \quad \mathbf{P}_{2j+1}^{(\ell)} := \mathbf{B}_{2j}^{(\ell)} \circ \mathbf{P}_j^{(\ell+1)}.$$

Throughout this recursive propagation, each node $\mathbf{P}_j^{(\ell)}$ precisely stores the cumulative product of propagators up to a specific index, preserving the invariant:

$$\mathbf{P}_j^{(\ell)} = \mathbf{U}_{j \cdot 2^\ell - 1} \circ \mathbf{U}_{j \cdot 2^\ell - 2} \circ \cdots \circ \mathbf{U}_0,$$

with the convention $\mathbf{U}_{-1} \equiv \mathrm{Id}_H$. This structured downsweep step efficiently leverages the hierarchical binary-tree structure to compute all prefix products in parallel, ensuring logarithmic computational depth and scalability.

*(III) Prefixes at leaves (Final step).* At the lowest level ($\ell = 0$), the exclusive prefix products directly yield the desired cumulative propagators:

$$\mathbf{P}_m^{(0)} = \mathbf{U}_{m-1} \circ \cdots \circ \mathbf{U}_0 = \Pi(m), \quad 1 \le m \le M'.$$

Hence, we obtain a highly efficient approximation for each discrete Lie-algebraic Koopman operator $\mathcal{K}_\theta^{t_m, s} \approx \mathbf{P}_m^{(0)}$ in (6). In summary, the presented balanced-tree prefix scan algorithm leverages Lie-group associativity to achieve parallel computation with a computational depth of $\mathcal{O}(\log M')$ and total computational complexity of $\mathcal{O}(M')$, ensuring scalability and practical feasibility for complex long-term dynamics. Fig 1 provides the conceptual illustration of a Lie-group valued prefix parallel scan algorithm.

While the balanced-tree prefix scan achieves an optimal *temporal* complexity of $\mathcal{O}(\log M')$ reduced from $\mathcal{O}(M)$, performing computations directly within the most ambient space $(G, \mathfrak{g}) := (GL(K), \mathfrak{gl}(K))$ yields no analogous reduction in *spatial* complexity. Specifically, each hierarchical merge remains computationally intensive, involving dense Lie group operations with cubic arithmetic complexity $\mathcal{O}(d^3)$. In what follows, we address these spatial complexity

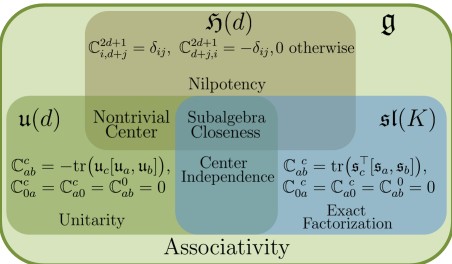

*Figure 2.* Summarization of Lie subalgebras and their properties.

issues by restricting computations to structured Lie subalgebras $\mathfrak{h} \subseteq \mathfrak{g}$, whose algebraic properties naturally enable computationally more efficient representations.

**Subalgebra Exploration.** Since our Lie-algebraic representation of the Koopman operator depends explicitly and exclusively on the structure constants $\mathbb{C}_{ab}^c$, exploring different Lie subalgebras amounts to specifying these constants accordingly. Though there may exist some other suitable subalgebras, but to the best of our knowledge with considerations for implementation complexity, we investigate three canonical Lie subalgebras, explicitly characterizing their algebraic structures and respective structure constants as summarized in Fig 2.

**Heisenberg Algebra $\mathfrak{H}(d)$.** With the formal Lie algebraic framework established, we now introduce the *Heisenberg Lie algebra* as our first candidate subalgebra of $\mathfrak{g}$. This algebra allows significant computational simplifications while preserving essential algebraic structures. Consider the Lie subalgebra $\mathfrak{H}(d) \subseteq \mathfrak{g}$ defined explicitly by its $(2d + 1)$-dimensional basis $\mathfrak{H}(d) = \mathrm{span}\{\mathfrak{q}_1, \ldots, \mathfrak{q}_d, \mathfrak{p}_1, \ldots, \mathfrak{p}_d, \mathfrak{z}\}$. The algebraic structure of $\mathfrak{H}(d)$ is completely characterized by its nontrivial Lie brackets:

$$[\mathfrak{q}_i, \mathfrak{p}_j] = \delta_{ij}\mathfrak{z}, \quad [\mathfrak{q}_i, \mathfrak{q}_j] = [\mathfrak{p}_i, \mathfrak{p}_j] = [\mathfrak{z}, \cdot] = 0, \quad (10)$$

$$\mathrm{Ad}_{\mathfrak{H}(d)}^2(\mathfrak{H}(d)) = [\mathfrak{H}(d), [\mathfrak{H}(d), \mathfrak{H}(d)]] = 0, \quad (11)$$

Then, with the index convention $\mathfrak{h}_i = \mathfrak{q}_i$, $\mathfrak{h}_{d+i} = \mathfrak{p}_i$ and $\mathfrak{H}(d) = \mathfrak{z}$ for all $1 \le i \le d$, one has

$$\mathbb{C}_{i, d+j}^{2d+1} = \delta_{ij}, \quad \mathbb{C}_{d+j, i}^{2d+1} = -\delta_{ij}, \quad \mathbb{C}_{ab}^c = 0. \quad (12)$$

By construction in (11) and (12), the nilpotent property of the Heisenberg algebra guarantees that the Magnus series in Proposition 2.2 truncates exactly at *the second order*, reducing the infinite expansion in Prop 2.2 to only two terms ($m = 2$). Consequently, the higher-order commutator contributions vanish beyond the second level, substantially mitigating local truncation errors and yielding markedly improved accuracy relative to generic non-nilpotent settings.

**Unitary Algebra $\mathfrak{u}(d)$.** Motivated by symmetry preservation in closed, finite-dimensional quantum dynamics (Stone, 1932; Ismael, 2024; Nielsen & Chuang, 2010), we adopt the

unitary Lie subalgebra $\mathfrak{u}(d) \subseteq \mathfrak{g}$ as our second candidate:

$$\mathfrak{u}(d) = \{X \in \mathbb{C}^{d \times d} : X^\top = -X\},$$
$$\mathfrak{su}(d) = \{X \in \mathfrak{u}(d) : \operatorname{tr}(X) = 0\}.$$

For an orthonormal skew-Hermitian basis $\{\mathfrak{u}_a\}_{a=0}^{d^2-1}$ of $\mathfrak{u}(d)$ under $\langle X, Y \rangle = -\operatorname{tr}(XY)$, with the central element

$$\mathfrak{u}_0 = -iI/\sqrt{d}, \quad [\mathfrak{u}_0, \cdot] = 0, \quad \mathfrak{u}(d) = \mathfrak{su}(d) \oplus \mathbb{R}\mathfrak{u}_0. \quad (13)$$

With this ordered basis, the Lie bracket closes with the structure constants are given by

$$\mathbb{C}_{ab}^c = -\operatorname{tr}(\mathfrak{u}_c[\mathfrak{u}_a, \mathfrak{u}_b]), \quad \mathbb{C}_{0a}^c = \mathbb{C}_{a0}^c = \mathbb{C}_{ab}^0 = 0. \quad (14)$$

Following from (14) and Proposition 2.2, it immediately follows that any Magnus truncation $\Omega_\theta^{[r]}(t + \Delta t, t)$ remains within the Lie algebra $\mathfrak{u}(d)$, thus ensuring the unitarity of each propagator $U = \exp(\Omega_\theta^{[r]}) \in U(d)$ at every segment and throughout the balanced-tree scan. As a direct consequence, exact norm preservation is guaranteed.

**Projective Algebra $\mathfrak{sl}(K)$.** The numerically favorable decomposition of the Lie algebra $\mathfrak{gl}(K)$ into a central part and a traceless part naturally motivates us to define the corresponding projective Lie algebra by quotienting out the one-dimensional *center* $\mathfrak{z} = \{\lambda I : \lambda \in \mathbb{R}\}$, as $\mathfrak{pgl}(K) = \mathfrak{gl}(K)/\mathfrak{z}$. Via the projection $\pi : \mathfrak{gl}(K) \to \mathfrak{sl}(K)$ where $\pi(X) = X - K^{-1}\operatorname{Tr}(X)I$, one has an isomorphism of Lie algebras $\mathfrak{pgl}(K) \cong \mathfrak{sl}(K)$, since $\pi$ is a Lie algebra homomorphism with kernel $\mathfrak{z}$. Given an orthonormal traceless basis $\{\mathfrak{s}_a\}_{a=1}^{K^2-1}$ of $\mathfrak{sl}(K)$ with respect to the Frobenius inner product, we extend it by including the central element analogous to (13), forming a complete basis:

$$\mathfrak{s}_0 = I/\sqrt{K}, \quad [\mathfrak{s}_0, \cdot] = 0, \quad \mathfrak{gl}(K) = \mathfrak{sl}(K) \oplus \mathbb{R}\mathfrak{s}_0. \quad (15)$$

With this ordered basis, the Lie bracket closes and the structure constants are summarized as follows:

$$\mathbb{C}_{ab}^c = \operatorname{tr}(\mathfrak{s}_c^\top [\mathfrak{s}_a, \mathfrak{s}_b]), \quad \mathbb{C}_{0a}^c = \mathbb{C}_{a0}^c = \mathbb{C}_{ab}^0 = 0. \quad (16)$$

To elucidate the structural benefits of this algebra, we express the neural generator as $A_\theta(t) = \alpha_\theta(t)\mathfrak{s}_0 + \sum_{a=1}^{K^2-1} \alpha_a(t)\mathfrak{s}_a$ where $\alpha_\theta(t) = \frac{\operatorname{tr}(A_\theta(t))}{\sqrt{K}}$. Since $\mathfrak{s}_0$ is central, (16) and Proposition 2.2 imply an exact decoupling of the Magnus exponent:

$$\Omega_\theta^{[r]}(t + \Delta t, t) = \left( \int_t^{t+\Delta t} \alpha_\theta(\tau)d\tau \right) \mathfrak{s}_0 + S_\theta^{[r]}(t + \Delta t, t).$$

During the balanced-tree scan, the central scalar aggregates by prefix-sum independently from the traceless component in $\operatorname{SL}(K)$. This center–traceless split enhances numerical stability without affecting algebraic closure or observables.

We implement the proposed framework by encoding an input state into a latent state, evolving the corresponding latent trajectory under the proposed Lie-algebraic dynamics, and decoding it back to the original state space. Further implementation details are provided in App. C.

# 4. Related Work

**Koopman Dynamics.** Koopman-based models linearize nonlinear dynamics, yet they differ in how time propagation is handled under non-stationary settings and evolving system characteristics. Koopa (Liu et al., 2023) separates stationary and time-varying parts and advances both with shared linear operators. KoNODE (Bai & Ding, 2025) allows ODE parameters to evolve over time, enabling more flexible representations than fixed-coefficient dynamics. KoVAE (Naiman et al., 2024) introduces a latent linear state-space prior that supports continuous-time generation and irregular sampling within a Koopman-inspired formulation. KooNPro (Zheng et al., 2025) captures stochastic spectra using Neural Processes for local–global fusion in uncertainty-aware prediction settings. MamKO (LI et al., 2025) generates operators dynamically from streaming inputs via Mamba for real-time rollout and downstream control tasks.

**Generative Models.** Generative models have been explored as a framework for modeling dynamical systems, as they enable stochastic trajectory generation while incorporating geometric or physical constraints. EqMotion (Xu et al., 2023) preserves frame consistency through SE(3)-equivariant interactions. GeoTDM (Han et al., 2024) denoises over time using diffusion while enforcing symmetry. Equivariant graph-based neural ODEs such as SEGNO (Liu et al., 2024) improve generalization in long-term physical dynamics modeling by incorporating physical inductive biases and continuous motion laws. These models encode dynamics through operator shifts, latent priors, or geometric constraints, yet the tradeoff between stability and efficiency in deep composition for long-horizon prediction scenarios remains an open challenge. Such limitations become more pronounced when trajectory generation relies on deep iterative refinement or long sampling chains, which can significantly increase computational cost.

# 5. Experiment

We study the initial value problem on atomistic molecular dynamics systems, including linear alkanes and the small peptide system. Starting from an initial condition $x_0$, the model predicts the system's temporal evolution by generating a sequence of future states $\{x_1, \ldots, x_M\}$ over a finite prediction horizon. For comparison, we group prior approaches into two categories, where Koopman-based models offer fast inference but often suffer from limited expressivity and error accumulation, while probabilistic sequential

*Table 1.* **Performance Comparison on Met-Enkephalin.** We report inference and training time, best MSE and physical metrics including $C_\alpha$-RMSD, Rg MAE, Jensen–Shannon divergence of the Ramachandran $(\phi, \psi)$ distribution, and $C_\alpha$–$C_\alpha$ distance error.

| Method | Inference Time (s) | Train Time (s) | Best MSE ↓ | $C_\alpha$-RMSD (Å) | Rg MAE (Å) | JS$(\phi, \psi)$ Div. | C$\alpha$–C$\alpha$ Dist. (Å) |
|---|---|---|---|---|---|---|---|
| *(A) Koopman based models (Low Accuracy, High Efficiency)* | | | | | | | |
| Koopa (Liu et al., 2023) | $9.57\times10^{-1}$ | $8.87\times10^{-1}$ | $2.17\times10^{-1}$ | $3.01\times10^{0}$ | $3.26\times10^{0}$ | $6.18\times10^{-1}$ | $4.01\times10^{0}$ |
| KoVAE (Naiman et al., 2024) | $1.76\times10^{0}$ | $6.42\times10^{-2}$ | $9.12\times10^{-2}$ | $4.51\times10^{-1}$ | $1.56\times10^{-1}$ | $4.11\times10^{-2}$ | $1.44\times10^{-1}$ |
| KooNPro (Zheng et al., 2025) | $1.86\times10^{0}$ | $3.58\times10^{0}$ | $9.62\times10^{-1}$ | $3.82\times10^{-1}$ | $2.25\times10^{-1}$ | $5.56\times10^{-1}$ | $3.15\times10^{-1}$ |
| KoNODE (Bai & Ding, 2025) | $9.29\times10^{-1}$ | $4.74\times10^{0}$ | $1.11\times10^{0}$ | $3.68\times10^{1}$ | $2.65\times10^{2}$ | $6.91\times10^{-1}$ | $3.48\times10^{2}$ |
| MamKO (LI et al., 2025) | $9.77\times10^{-2}$ | $4.00\times10^{-1}$ | $9.84\times10^{-2}$ | $8.28\times10^{-1}$ | $3.64\times10^{0}$ | $2.11\times10^{-1}$ | $5.69\times10^{0}$ |
| *(B) Generative Model based Probabilistic Sequential Models (High Accuracy, Low Efficiency)* | | | | | | | |
| EqMotion (Xu et al., 2023) | $3.35\times10^{0}$ | $3.37\times10^{0}$ | $5.01\times10^{0}$ | $2.98\times10^{1}$ | $3.02\times10^{1}$ | $5.45\times10^{-1}$ | $3.60\times10^{1}$ |
| SEGNO (Liu et al., 2024) | $1.85\times10^{0}$ | $5.87\times10^{0}$ | $5.94\times10^{-1}$ | $2.98\times10^{1}$ | $1.17\times10^{1}$ | $6.80\times10^{-1}$ | $9.91\times10^{0}$ |
| GeoTDM (Han et al., 2024) | $1.38\times10^{2}$ | $3.97\times10^{-1}$ | $4.38\times10^{-1}$ | $6.16\times10^{-1}$ | $1.40\times10^{-1}$ | $4.52\times10^{-1}$ | $3.43\times10^{-1}$ |
| Ours (Heisenberg) $\mathfrak{H}(d)$ | $\mathbf{6.36\times10^{-3}}$ | $1.94\times10^{-2}$ | $\mathbf{7.54\times10^{-2}}$ | $2.89\times10^{-1}$ | $\mathbf{4.59\times10^{-2}}$ | $6.79\times10^{-2}$ | $3.59\times10^{-2}$ |
| Ours (Unitary) $\mathfrak{u}(d)$ | $7.80\times10^{-3}$ | $\mathbf{1.85\times10^{-2}}$ | $1.13\times10^{-1}$ | $\mathbf{2.77\times10^{-1}}$ | $5.04\times10^{-2}$ | $8.18\times10^{-2}$ | $4.48\times10^{-2}$ |
| Ours (Projective) $\mathfrak{sl}(K)$ | $8.95\times10^{-3}$ | $1.85\times10^{-2}$ | $8.72\times10^{-2}$ | $2.88\times10^{-1}$ | $4.71\times10^{-2}$ | $1.03\times10^{-1}$ | $\mathbf{3.58\times10^{-2}}$ |

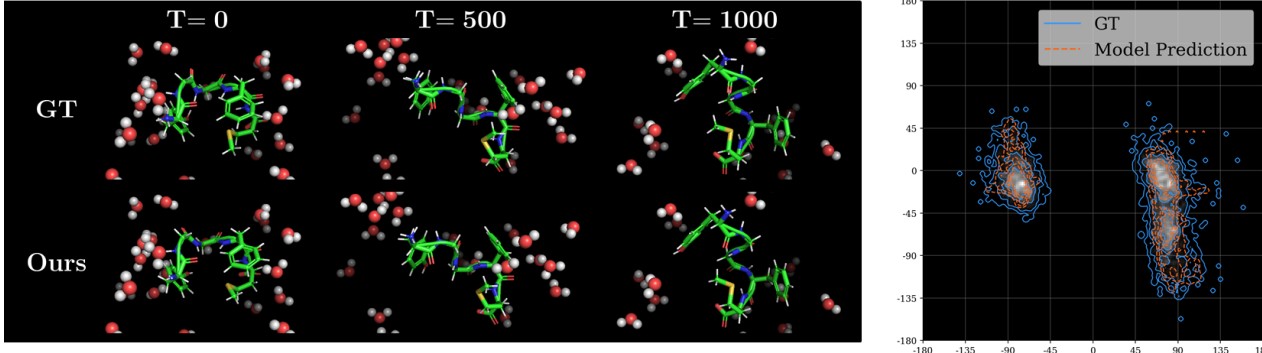

*Figure 3.* **Visualization of Met-Enkephalin.** *(Left)* Qualitative comparison of Met-Enkephalin on initial, midpoint and terminal state. *(Right)* Ramachandran map of Met-Enkephalin between ground-truth (GT) and model prediction.

generative models achieve higher accuracy at substantial computational cost.

**Peptide Structural Dynamics: Met-Enkephalin.** We perform molecular dynamics simulations of Met-Enkephalin, a canonical pentapeptide frequently employed as a benchmark system in biomolecular dynamics. To investigate peptide dynamics on a rugged free-energy landscape featuring multiple metastable conformational states and slow interconversion between distinct structural motifs, Met-enkephalin has been extensively used as a benchmark system for evaluating peptide folding behavior, enhanced sampling methodologies, and data-driven molecular dynamics models (Banerjee & Cukier, 2014). In our experiments, peptide dynamics are simulated under a classical Hamiltonian formalism using the AMBER ff14SB force field (Maier et al., 2015).

We evaluate model performance on Met-Enkephalin over the full trajectory length of $M = 1000$. As reported in Tab. 1, our three variants achieve superior performance over baseline models in terms of both accuracy and efficiency. In terms of computational efficiency, our methods achieve inference speeds of up to $2.2 \times 10^4$ times faster than prior approaches, while remaining more than two orders of magnitude faster than all baseline methods. Across all methods, substantial reductions in training time are also

observed, with wall-clock time reduced by up to two orders of magnitude compared to previous approaches. In terms of accuracy, our models attain the lowest mean squared error among all methods, with best MSE of $7.54 \times 10^{-2}$, where best MSE denotes the average frame-wise MSE over the full trajectory. We confirm strong physical fidelity, as evidenced by $C_\alpha$-RMSD values below $3 \times 10^{-1}$ Å. Under additional physical metrics, consistent structural agreement is observed, including accurate radii of gyration reflecting global compactness, low Jensen–Shannon divergences in backbone torsional distributions, and reduced long-range structural deviations as measured by $C_\alpha$–$C_\alpha$ distance errors. As shown in Fig. 3, qualitative visualizations further corroborate these findings by demonstrating close agreement between predicted and reference peptide conformations as well as their corresponding Ramachandran distributions, which characterize the backbone torsional conformations explored over the trajectory.

**Hydrocarbon Chain Dynamics: Alkanes.** To further validate generalizability and performance of our methods, we conduct molecular dynamics simulations of the alkane family, ranging from ethane ($C_2H_6$) to nonane ($C_9H_{20}$). These alkane systems serve as widely adopted benchmarks due to their well-characterized molecular complexity and broad relevance in chemical and biological research, such as hy-

*Table 2.* **Performance Comparison on Alkane Molecular Dynamics.** Performance comparison across different scales of alkane molecules grouped by complexity. We report average validation MSE, inference time efficiency, and physical fidelity for each model.

| Method | Short Chain ($C_2 - C_3$) | | Medium Chain ($C_4 - C_6$) | | Long Chain ($C_7 - C_9$) | | Physical Metric |
| --- | --- | --- | --- | --- | --- | --- | --- |
| | Best MSE ↓ | Inf time(s) | Best MSE ↓ | Inf time(s) | Best MSE ↓ | Inf time(s) | RMSD (Å) ↓ |
| *(A) Koopman Theory based Sequential Models* (Low Accuracy, High Efficiency) | | | | | | | |
| Koopa (Liu et al., 2023) | $9.06 \times 10^{-1}$ | $3.18 \times 10^{-2}$ | $8.77 \times 10^{-1}$ | $3.91 \times 10^{-2}$ | $5.40 \times 10^{-1}$ | $7.43 \times 10^{-2}$ | $4.45 \times 10^{-1}$ |
| KoVAE (Naiman et al., 2024) | $\mathbf{5.72 \times 10^{-2}}$ | $8.94 \times 10^{-2}$ | $1.22 \times 10^{-1}$ | $1.16 \times 10^{-1}$ | $1.97 \times 10^{-1}$ | $1.25 \times 10^{-1}$ | $4.42 \times 10^{-1}$ |
| KoonPro (Zheng et al., 2025) | $1.28 \times 10^{1}$ | $4.31 \times 10^{0}$ | $6.74 \times 10^{0}$ | $3.95 \times 10^{0}$ | $7.49 \times 10^{1}$ | $4.83 \times 10^{0}$ | $4.86 \times 10^{-1}$ |
| KoNODE (Bai & Ding, 2025) | $6.04 \times 10^{-1}$ | $7.26 \times 10^{0}$ | $7.03 \times 10^{-1}$ | $7.12 \times 10^{0}$ | $2.88 \times 10^{-1}$ | $7.35 \times 10^{0}$ | $1.06 \times 10^{0}$ |
| MamKO (LI et al., 2025) | $9.89 \times 10^{-1}$ | $8.75 \times 10^{-1}$ | $7.34 \times 10^{-1}$ | $8.48 \times 10^{-1}$ | $6.83 \times 10^{-1}$ | $8.92 \times 10^{-1}$ | $5.76 \times 10^{-1}$ |
| *(B) Generative Model based Probabilistic Sequential Models* (High Accuracy, Low Efficiency) | | | | | | | |
| EqMotion (Xu et al., 2023) | $1.62 \times 10^{-1}$ | $2.96 \times 10^{0}$ | $1.01 \times 10^{-1}$ | $3.12 \times 10^{0}$ | $1.38 \times 10^{-1}$ | $3.97 \times 10^{0}$ | $5.76 \times 10^{-1}$ |
| SEGNO (Liu et al., 2024) | $5.74 \times 10^{-1}$ | $1.78 \times 10^{0}$ | $5.06 \times 10^{-1}$ | $1.79 \times 10^{0}$ | $4.45 \times 10^{-1}$ | $1.81 \times 10^{0}$ | $2.83 \times 10^{-1}$ |
| GeoTDM (Han et al., 2024) | $6.46 \times 10^{-2}$ | $3.26 \times 10^{1}$ | $5.87 \times 10^{-2}$ | $7.08 \times 10^{1}$ | $5.41 \times 10^{-2}$ | $9.52 \times 10^{1}$ | $5.81 \times 10^{-1}$ |
| Ours (Heisenberg) $\mathfrak{H}(d)$ | $7.27 \times 10^{-2}$ | $2.68 \times 10^{-2}$ | $4.55 \times 10^{-2}$ | $\mathbf{2.84 \times 10^{-2}}$ | $4.93 \times 10^{-2}$ | $\mathbf{2.75 \times 10^{-2}}$ | $1.48 \times 10^{-1}$ |
| Ours (Unitary) $\mathfrak{u}(d)$ | $1.21 \times 10^{-1}$ | $4.75 \times 10^{-2}$ | $5.52 \times 10^{-2}$ | $5.38 \times 10^{-2}$ | $6.49 \times 10^{-2}$ | $5.13 \times 10^{-2}$ | $1.20 \times 10^{-1}$ |
| Ours (Projective) $\mathfrak{sl}(K)$ | $7.33 \times 10^{-2}$ | $\mathbf{2.57 \times 10^{-2}}$ | $\mathbf{4.31 \times 10^{-2}}$ | $3.12 \times 10^{-2}$ | $4.66 \times 10^{-2}$ | $2.99 \times 10^{-2}$ | $\mathbf{1.18 \times 10^{-1}}$ |

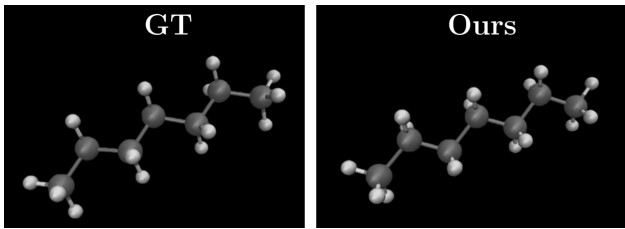

*Figure 4.* Qualitative comparison on molecular dynamics of heptane ($C_7H_{16}$) on T = 1000.

drocarbon–water interfaces in petroleum engineering (Xu et al., 2022). The alkane family constitutes a well-studied and systematically scalable family of hydrocarbons, where increasing the alkane chain length introduces additional carbon atoms and internal degrees of freedom, progressively raising the complexity of molecular interactions. In this context, we employ the *General AMBER Force Field* to define the underlying Hamiltonian dynamics governing the molecular simulations in Eq. (17).

We examine the model performance at the whole timestep of $M = 1000$. As shown in Tab 2, our method attains state-of-the-art accuracy while maintaining short inference time overall. Relative to the strongest prior, inference is faster by approximately $2.3 \times 10^3 - 3.2 \times 10^3$ times, with lower error in both cases. Although KoVAE model performs better in one case, our method achieves about $3.5 \times$ lower latency at similar $10^{-2}$ s performance scale. Overall, strong baselines with competitive accuracy require orders of magnitude more time on the medium and long groups, whereas our method consistently operates in a low-latency regime. For qualitative assessment, we visualize predictions on heptane with atom coordinates for timestep 1000 in Fig 4.

**Ablation Study.** We conduct an ablation study on the Lotka–Volterra system $\dot{x}(t) = \alpha x(t) - \beta x(t)y(t)$, $\dot{y}(t) = \delta x(t)y(t) - \gamma y(t)$, to evaluate inference-time scaling behavior and computational efficiency under increasing tempo-

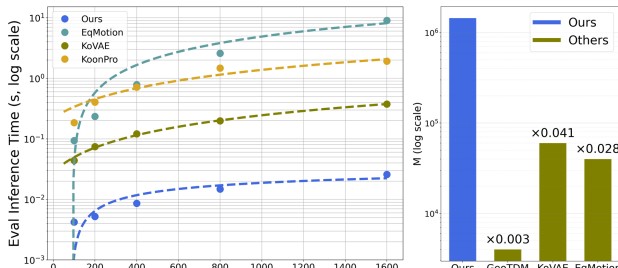

*Figure 5.* Ablation Study for Timestep Increasing Rate (Left) and Maximum Timestep Scalability (Right).

ral horizons. Despite its low-dimensional formulation, the Lotka–Volterra system characterized by oscillatory trajectories and long-range temporal dependencies, thus accurately predicting its dynamics over extended horizons remains challenging due to the accumulation of nonlinear interactions over time. Specifically, inference time is measured while doubling the number of timesteps from $M = 100$ to $M = 1600$. As shown in the left panel of Fig. 5, our model achieves a speedup ranging from $\mathbf{14.5 \times}$ to $\mathbf{348 \times}$ at $M = 1600$ relative to baseline methods. Although the temporal horizon is increased by a factor of 16, the runtime of our method grows by only $6.13 \times$, indicating a substantially improved scaling efficiency compared to other models as the sequence length increases. This stands in contrast to baseline methods, whose inference cost increases proportionally with the rollout length. In addition, we evaluate the maximum temporal horizon that each model can handle on the same Lotka–Volterra system. Using a single GPU, baseline methods fail to learn long-range dynamics due to memory limitations or insufficient model capacity. In contrast, our model successfully learns sequences of up to $\mathbf{1.45 \times 10^6}$ timesteps, which successfully demonstrates efficiency for extremely long horizons. More details are provided in App. D.

# 6. Conclusion

In this work, we present a Lie-algebraic framework for modeling Koopman dynamics via infinitesimal generators and the Magnus expansion, offering a principled and scalable approach to learning dynamics. By parameterizing within finite-dimensional Lie subalgebras and leveraging their structure constants, the model maintains algebraic consistency while enabling efficient time-ordered integration. By using the associativity property, we propose a parallel scan algorithm over Lie group elements, reducing temporal complexity to $\mathcal{O}(\log M)$ without structure collapse. Empirically, our method shows strong time efficiency and long term stability, with acceleration along the temporal axis.

## Impact Statement

This paper presents work whose goal is to advance the field of machine learning. There are many potential societal consequences of our work, none of which we feel must be specifically highlighted here.

## Acknowledgments

This work was supported by the Institute of Information & Communications Technology Planning & Evaluation(IITP)-ICT Creative Consilience Program grant funded by the Korea government(MSIT)(IITP-2026-RS-2020-II201819). This work was also supported by the National Research Foundation of Korea(NRF) grant funded by the Korea government(MSIT)(RS-2026-25495269). This research was supported by Basic Science Research Program through the NRF funded by the Korea government(MSIT)(RS-2026-25522655). Additional support was provided by Korea University Faculty Start-up Research Grant(K2506311). Further support was provided by Korea University Faculty Start-up Research Grant(K2506541).

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

# A. Preliminary

## A.1. Lie groups, Lie algebras, and structure constants

This subsection summarizes the algebraic notation used throughout the paper. We restrict to finite-dimensional matrix Lie groups. A matrix Lie group is a smooth subgroup $G \subset \mathrm{GL}(D)$ equipped with matrix multiplication and inversion. Its associated Lie algebra is $\mathfrak{g} := T_{I_D}G \subset \mathbb{R}^{D \times D}$, which is closed under the commutator $[X, Y] := XY - YX$.

Let $\mathfrak{h} \subseteq \mathfrak{g}$ be a Lie subalgebra of dimension $B$. We fix a basis $\{\mathfrak{h}_a\}_{a=1}^B$ of $\mathfrak{h}$ and represent elements of $\mathfrak{h}$ in coefficient form. Specifically, any $A \in \mathfrak{h}$ is written as $A = \sum_{a=1}^B \alpha_a \mathfrak{h}_a$, $\alpha = (\alpha_1, \ldots, \alpha_B)^\top \in \mathbb{R}^B$. The Lie bracket in $\mathfrak{h}$ closes in this basis, $[\mathfrak{h}_a, \mathfrak{h}_b] = \sum_{c=1}^B \mathbb{C}_{ab}^c \mathfrak{h}_c$, which defines the structure constants $\mathbb{C} \in \mathbb{R}^{B \times B \times B}$. These constants encode the algebraic relations among the basis elements and determine the form of all nested commutators.

We introduce the quantities $\mathrm{nnz}(\mathbb{C}) := \big|\{(a,b,c) : \mathbb{C}_{ab}^c \neq 0\}\big|$, $\Delta_\mathbb{C} := \max_b \big|\{(a,c) : \mathbb{C}_{ab}^c \neq 0\}\big|$, which summarize the sparsity pattern of the structure constants. Several Lie algebras considered in this paper admit a decomposition $\mathfrak{h} = \mathbb{R}\mathfrak{z} \oplus \mathfrak{s}$, where $\mathbb{R}\mathfrak{z}$ is a one-dimensional ideal and $\mathfrak{s}$ is a complementary subalgebra, that separates a central direction from the remaining noncentral components.

## A.2. Magnus expansion

This subsection recalls the Magnus expansion as a standard representation of linear time-dependent evolutions. We fix notation and conventions used throughout the paper, in particular the segmentwise formulation adopted in the construction of discrete Koopman propagators.

Let $A : [s, t] \to \mathbb{R}^{D \times D}$ be a (piecewise) continuous matrix-valued function. Consider the linear initial value problem $\frac{\mathrm{d}}{\mathrm{d}\tau} U(\tau, s) = A(\tau) U(\tau, s)$, $U(s, s) = I_D$, where $U(\tau, s) \in \mathrm{GL}(D)$ denotes the propagator from time $s$ to $\tau$. Instead of expressing $U(t, s)$ as a time-ordered product, the Magnus expansion represents the solution as a single matrix exponential $U(t, s) = \exp\big(\Omega(t, s)\big)$, where the exponent $\Omega(t, s) \in \mathfrak{gl}(D)$ is given by the series $\Omega(t, s) = \sum_{m=1}^\infty \Omega^{(m)}(t, s)$. Each term $\Omega^{(m)}$ consists of a time-ordered integral of depth $m$ whose integrand is formed by nested commutators of the generator $A(\cdot)$. The first two terms are $\Omega^{(1)}(t, s) = \int_s^t A(\tau_1) \, \mathrm{d}\tau_1$, $\Omega^{(2)}(t, s) = \frac{1}{2} \int_s^t \int_s^{\tau_1} [A(\tau_1), A(\tau_2)] \, \mathrm{d}\tau_2 \, \mathrm{d}\tau_1$, and the higher-order terms involve iterated commutators integrated over the corresponding time-ordered simplices $s \leq \tau_m \leq \cdots \leq \tau_1 \leq t$.

Throughout this paper the generator is constrained to a finite-dimensional Lie subalgebra $\mathfrak{h} \subset \mathfrak{gl}(D)$ with basis $\{\mathfrak{h}_a\}_{a=1}^B$. Writing $A(\tau) = \sum_{a=1}^B \alpha_a(\tau) \mathfrak{h}_a$, $\Omega^{(m)}(t, s) = \sum_{c=1}^B \omega^{(m),c}(t, s) \mathfrak{h}_c$, the Magnus expansion reduces to scalar combinations of the coefficients $\alpha_a(\tau)$ weighted by the structure constants of $\mathfrak{h}$. In particular, the first-order coefficients are given by $\omega^{(1),c}(t, s) = \int_s^t \alpha_c(\tau) \, \mathrm{d}\tau$. For numerical purposes we apply the Magnus construction on each segment $[t_k, t_{k+1}]$ of a partition of the time interval, with the segmentwise truncated exponent is defined as $\Omega_k^{[p]} := \sum_{m=1}^p \Omega^{(m)}(t_{k+1}, t_k)$, and the corresponding one-step propagator is $U_k^{[p]} := \exp\big(\Omega_k^{[p]}\big)$.

## A.3. Algebraic Composition and Prefix-Scan Merges

In this subsection we formalize the Lie-algebraic combine operation used by the prefix-scan algorithm. Let $\mathfrak{h} \subset \mathfrak{gl}(D)$ be a finite-dimensional Lie algebra with basis $\{\mathfrak{h}_a\}_{a=1}^B$ and structure constants $[\mathfrak{h}_a, \mathfrak{h}_b] = \sum_{c=1}^B \mathbb{C}_{ab}^c \mathfrak{h}_c$. We denote by $H \subset \mathrm{GL}(D)$ the connected Lie group generated by $\mathfrak{h}$, equipped with the exponential map $\exp : \mathfrak{h} \to H$.

Given $X, Y \in \mathfrak{h}$, we define the *algebraic composition* map $\mathrm{Comp}(X, Y) \in \mathfrak{h}$ implicitly by the identity

$$\exp(X) \exp(Y) = \exp\big(\mathrm{Comp}(X, Y)\big).$$

The element $\mathrm{Comp}(X, Y)$ admits the Lie-series expansion

$$\mathrm{Comp}(X, Y) = X + Y + \tfrac{1}{2}[X, Y] + \tfrac{1}{12}\big([X, [X, Y]] + [Y, [Y, X]]\big) + \mathcal{O}(\|X\|^4 + \|Y\|^4).$$

More generally, $\mathrm{Comp}(X, Y)$ is given by an infinite sum of nested commutators in $X$ and $Y$ whose coefficients depend only on Bernoulli numbers. In coefficient form, writing $X = \sum_{a=1}^B \omega^a \mathfrak{h}_a$, $Y = \sum_{b=1}^B \eta^b \mathfrak{h}_b$, the Lie bracket is $[X, Y] = \sum_{a,b,c} \omega^a \eta^b \mathbb{C}_{ab}^c \mathfrak{h}_c$, and higher nested commutators are multivariate polynomials in the coefficients $\{\omega^a\}$ and $\{\eta^b\}$ with coefficients determined by the structure constants $\mathbb{C}_{ab}^c$. In the segmentwise Magnus construction, each time step $[t_k, t_{k+1}]$ yields a Lie-algebra increment $\Omega_k \in \mathfrak{h}$ and an associated propagator $U_k = \exp(\Omega_k) \in H$. Composition of two adjacent

segments corresponds to the Lie-algebraic merge

$$\Omega_{\mathrm{merge}} := \mathrm{Comp}(\Omega_{k+1}, \Omega_k), \qquad \exp(\Omega_{\mathrm{merge}}) = U_{k+1} U_k.$$

For numerical purposes, the composition series is truncated at order $p \geq 1$, yielding a truncated merge operator

$$\mathrm{merge}_p(X, Y) := \mathrm{Comp}_p(X, Y),$$

where $\mathrm{Comp}_p$ retains all commutator terms up to order $p$. The exact composition map satisfies associativity through the group product,

$$\exp\big(\mathrm{Comp}(X, Y)\big) = \exp(X) \exp(Y),$$

and therefore inherits associativity from multiplication in $H$. The truncated map $\mathrm{Comp}_p$ is associative up to terms of order $p + 1$ in the Lie bracket. Given a sequence of Lie-algebra increments $\{\Omega_k\}_{k=0}^{M-1} \subset \mathfrak{h}$, the discrete-time propagators $U_k = \exp(\Omega_k)$ define cumulative products

$$U_{[0,m)} := U_{m-1} \cdots U_0, \qquad 1 \leq m \leq M.$$

A balanced-tree prefix scan computes all $U_{[0,m)}$ using repeated pairwise merge operations. At the group level, these merges are realized by multiplication in $H$, and they correspond to repeated applications of the truncated composition merge at the Lie-algebra level.

### A.4. Subalgebra Families

We summarize the algebraic structure of the Lie subalgebras in the main text defined by a choice of basis and structure constants, which determine the form of the Magnus expansion. The $d$-dimensional Heisenberg algebra is the $(2d + 1)$-dimensional Lie algebra $\mathfrak{H}(d) = \mathrm{span}\{\mathfrak{q}_1, \ldots, \mathfrak{q}_d, \mathfrak{p}_1, \ldots, \mathfrak{p}_d, \mathfrak{z}\}$, with Lie brackets $[\mathfrak{q}_i, \mathfrak{p}_j] = \delta_{ij}\, \mathfrak{z}$, $[\mathfrak{p}_j, \mathfrak{q}_i] = -\delta_{ij}\, \mathfrak{z}$, and all other brackets equal to zero. Indexing the basis as $\mathfrak{h}_i = \mathfrak{q}_i$, $\mathfrak{h}_{d+i} = \mathfrak{p}_i$, $\mathfrak{h}_{2d+1} = \mathfrak{z}$, the only nonzero structure constants are $\mathbb{C}_{i,\, d+j}^{2d+1} = \delta_{ij}$, $\mathbb{C}_{d+j,\, i}^{2d+1} = -\delta_{ij}$. The lower central series satisfies $[\mathfrak{H}(d), \mathfrak{H}(d)] = \mathbb{R}\mathfrak{z}$, $[\mathfrak{H}(d), \mathbb{R}\mathfrak{z}] = 0$, so $\mathfrak{H}(d)$ is step-2 nilpotent. Consequently, all nested commutators of depth greater than one vanish, and $\Omega^{(m)} \equiv 0$ for $m \geq 3$ in the Magnus expansion. The BCH formula reduces to $\mathrm{BCH}(X, Y) = X + Y + \frac{1}{2}[X, Y]$.

The unitary Lie algebra is defined as $\mathfrak{u}(D) = \{A \in \mathbb{C}^{D \times D} : A^* = -A\}$, with Lie bracket $[A, B] = AB - BA$. It decomposes as $\mathfrak{u}(D) = \mathfrak{su}(D) \oplus \mathbb{R}\mathfrak{u}_0$, $\mathfrak{u}_0 = -\frac{i}{\sqrt{D}}I$, where $\mathfrak{u}_0$ spans the center. The subspace $\mathfrak{u}(D)$ is closed under commutators, and if $A(t) \in \mathfrak{u}(D)$ for all $t$, then all Magnus terms $\Omega^{(m)}(t, s)$ also lie in $\mathfrak{u}(D)$. Therefore $\exp(\Omega(t, s))$ is unitary for every truncation order.

For the projective algebra $\mathfrak{sl}(D)$, the Lie algebra $\mathfrak{gl}(D)$ admits the decomposition $\mathfrak{gl}(D) = \mathbb{R}I \oplus \mathfrak{sl}(D)$, $\mathfrak{sl}(D) = \{A \in \mathbb{R}^{D \times D} : \mathrm{tr}(A) = 0\}$. Since the identity matrix $I$ is central, all commutators depend only on the traceless components, $[A, B] = [A - \frac{1}{D}\mathrm{tr}(A)I,\, B - \frac{1}{D}\mathrm{tr}(B)I]$. Writing a generator as $A(t) = \alpha(t)I + S(t)$ with $S(t) \in \mathfrak{sl}(D)$, the Magnus expansion decomposes as $\Omega(t, s) = \left(\int_s^t \alpha(\tau)\, d\tau\right)I + \sum_{m \geq 1} \Omega_{\mathfrak{sl}}^{(m)}(t, s)$, $\Omega_{\mathfrak{sl}}^{(m)}(t, s) \in \mathfrak{sl}(D)$. The scalar component aggregates additively, while all nontrivial commutator structure is confined to $\mathfrak{sl}(D)$.

## B. Theoretical Results

In this section, we formalize the computational complexity analysis underlying the Lie-algebraic Koopman framework and the parallel composition scheme introduced in the main text. The goal of this section is to make explicit the cost of constructing local Magnus expansions on segmented time intervals and of composing the resulting propagators via a balanced prefix scan on the Lie group. Based on this model, we characterize the total work, parallel depth, and memory complexity for a time horizon partitioned into $M$ segments.

For complexity analysis, we distinguish two representations. In the coefficient mode, Lie elements are represented by their coordinates in a fixed basis of size $B$ while the matrix mode, the same elements are realized in a $D \times D$ matrix representation. We also consider a general $Q$-point quadrature per segment to keep the Magnus leaf construction fully general.

Let a time interval be partitioned into $M$ segments with $Q$ quadrature nodes per segment. We denote by $B$ the Lie basis size, by $D$ the matrix dimension of the chosen representation, and by $\mathbb{C}_{ab}^c$ the structure constants of the subalgebra. Write

$\mathrm{nnz}(\mathbb{C}) := |\{(a,b,c) : \mathbb{C}_{ab}^{\,c} \neq 0\}|$ and $\Delta_{\mathbb{C}} := \max_b |\{(a,c) : \mathbb{C}_{ab}^{\,c} \neq 0\}|$. Balanced prefix scan over $M$ leaves has work $O(M)$ combines and depth $O(\log M)$. Firstly quantify the cost of constructing a single Magnus leaf under a general quadrature scheme, before analyzing the complexity of the scan itself.

---

**Proposition B.1** (Leaf cost). *With a truncation order $p \geq 1$, the cost to form $\Omega^{[p]}(t_{k+1}, t_k) = \sum_{m=1}^{p} \Omega^{(m)}(t_k, \Delta t_k)$ in the* coefficient mode *is bounded by*

$$\mathcal{C}_{\mathrm{leaf}}(p) = O(QB) \;+\; O\big(Q^2 \mathrm{nnz}(\mathbb{C})\big) \;+\; \sum_{m=3}^{p} O\big(Q^m B \Delta_{\mathbb{C}}^{m-1}\big).$$

---

*Proof.* Fix a segment $[t_k, t_{k+1}]$ and write $A(\tau) = \sum_{a=1}^{B} \alpha_a(\tau) \, \mathfrak{h}_a$ and $\Omega^{(m)}(t_{k+1}, t_k) = \sum_{c=1}^{B} \omega^{(m),c} \, \mathfrak{h}_c$. All time-ordered integrals are evaluated by a $Q$-point quadrature rule per segment. Up to constant factors, an $m$-fold time-ordered integral gives $\Theta(Q^m)$ quadrature samples.

The first-order term $(m=1)$ $\omega^{(1),c} = \int_{t_k}^{t_{k+1}} \alpha_c(\tau) \, d\tau$ is computed as a weighted sum over $Q$ nodes. This requires $B$ dot products of length $Q$, hence

$$\mathcal{C}_{\mathrm{leaf}}^{(1)} = O(QB).$$

For $m = 2$, the integrand is $[A(\tau_1), A(\tau_2)]$. Writing $x^a = \alpha_a(\tau_1)$ and $y^b = \alpha_b(\tau_2)$, $[x, y]^c = \sum_{a,b} x^a y^b \, \mathbb{C}_{ab}^{\,c}$. We evaluate the bracket by iterating over the nonzero triples $(a, b, c)$ of $\mathbb{C}$, so that one bracket evaluation costs $O(\mathrm{nnz}(\mathbb{C}))$. Since the quadrature of the time-ordered double integral uses $\Theta(Q^2)$ samples,

$$\mathcal{C}_{\mathrm{leaf}}^{(2)} = O(Q^2 \, \mathrm{nnz}(\mathbb{C})).$$

In the case of $m \geq 3$, each $\Omega^{(m)}$ consists of $m$-fold time-ordered integrals whose integrands are nested commutators of depth $m - 1$. We bound the cost of evaluating a generic nested commutator $\mathrm{ad}_{v_1} \mathrm{ad}_{v_2} \cdots \mathrm{ad}_{v_{m-1}}(v_m)$, $v_i \in \mathbb{R}^B$. Fixing an index $b$, the set $\{(a, c) : \mathbb{C}_{ab}^{\,c} \neq 0\}$ has cardinality at most $\Delta_{\mathbb{C}}$. Evaluating $\mathrm{ad}_v(\cdot) = [v, \cdot]$ in coefficient mode can therefore increase the number of contributing terms by at most a factor $\Delta_{\mathbb{C}}$. After $m - 1$ nested adjoint applications, the cost of forming all $B$ output coefficients is bounded by $O(B \, \Delta_{\mathbb{C}}^{m-1})$. Combining this with $\Theta(Q^m)$ quadrature samples yields

$$\mathcal{C}_{\mathrm{leaf}}^{(m)} = O\big(Q^m B \, \Delta_{\mathbb{C}}^{m-1}\big), \qquad m \geq 3.$$

Summing the contributions for $m = 1, \ldots, p$ gives $\mathcal{C}_{\mathrm{leaf}}(p) = O(QB) + O(Q^2 \, \mathrm{nnz}(\mathbb{C})) + \sum_{m=3}^{p} O\big(Q^m B \, \Delta_{\mathbb{C}}^{m-1}\big)$. $\quad\square$

---

**Proposition B.2** (Scan cost and memory). *Let $\mathcal{C}_{\mathrm{merge}}$ be the per-node group combine cost in the chosen representation. Then the total work and parallel depth for a horizon of $M$ segments satisfy $\mathcal{W} = M \cdot \mathcal{C}_{\mathrm{leaf}}(p) + (M - 1) \cdot \mathcal{C}_{\mathrm{merge}}$, $\mathcal{D} = O(\log M)$. Naive storage is $O(MB)$ in coefficient mode, or $O(MD^2)$ in matrix mode. With on-the-fly merges and level buffering, it reduces to $O(B \log M)$ or $O(D^2 \log M)$ respectively.*

---

*Proof.* A horizon partitioned into $M$ segments produces $M$ independent Magnus leaves $\{\Omega_k^{[p]}\}_{k=0}^{M-1}$. By definition, forming each leaf costs $\mathcal{C}_{\mathrm{leaf}}(p)$, hence the total leaf work is $M \, \mathcal{C}_{\mathrm{leaf}}(p)$.

The balanced scan combines leaves by a full binary tree of pairwise merges. A binary tree with $M$ leaves has exactly $M - 1$ internal nodes, and each internal node performs one merge. If the per-merge cost is $\mathcal{C}_{\mathrm{merge}}$, the total merge work is $(M - 1)\mathcal{C}_{\mathrm{merge}}$. Therefore

$$\mathcal{W} = M \, \mathcal{C}_{\mathrm{leaf}}(p) + (M - 1) \, \mathcal{C}_{\mathrm{merge}}.$$

For parallel depth, the balanced tree has height $\lceil \log_2 M \rceil$, and all merges at the same level are independent and can be executed in parallel. Thus the scan depth satisfies $\mathcal{D} = O(\log M)$. For memory, storing all $M$ leaf (or prefix) Lie-algebra elements requires $O(MB)$ coefficients in coefficient mode, or $O(MD^2)$ entries in matrix mode. With level buffering, one stores only the intermediate partial products needed at each tree level; at any time there are $O(\log M)$ active levels, and each stored element costs $O(B)$ (coefficient mode) or $O(D^2)$ (matrix mode). Hence the peak memory reduces to $O(B \log M)$ or $O(D^2 \log M)$, respectively. $\quad\square$

With the leaf construction cost established, we next analyze the total work, parallel depth, and memory complexity induced by the balanced scan. For completeness, we summarize the dominant per-merge costs in several common representations used throughout the paper. We summarize the dominant per-merge costs for several common representations. Matrix product on $D \times D$ uses $\mathcal{C}_{\mathrm{merge}} = O(D^3)$. Heisenberg group law admits $O(B)$ combine. Projective split adds an $O(1)$ scalar channel merge decoupled from the traceless product.

We now instantiate the above with the algebraic properties of each subalgebra. In each case, the bounds follow directly from the structure constants and the resulting truncation behavior of the Magnus and BCH expansions.

---

**Theorem B.3** (Heisenberg $\mathfrak{H}(d)$). *For the Heisenberg algebra with structure constants in* (12)*, one has step-2 nilpotency, hence* $\Omega^{(m)} \equiv 0$ *for* $m \geq 3$*. Moreover* $\mathrm{nnz}(\mathbb{C}) = 2d$ *and* $\Delta_{\mathbb{C}} = 1$*. Therefore*

$$\mathcal{C}_{\mathrm{leaf}}^{\mathrm{H}} = O(Qd) + O(Q^2 d), \qquad \mathcal{C}_{\mathrm{merge}}^{\mathrm{H}} = O(d).$$

*The total work and depth are*

$$\mathcal{W}_{\mathrm{H}} = O\big(Md(Q + Q^2 + 1)\big), \qquad \mathcal{D}_{\mathrm{H}} = O(\log M),$$

*with memory* $O(Md)$ *(or* $O(d \log M)$ *with level buffering).*

---

*Proof.* We work with $\mathfrak{H}(d) = \mathrm{span}\{\mathfrak{q}_1, \ldots, \mathfrak{q}_d, \mathfrak{p}_1, \ldots, \mathfrak{p}_d, \mathfrak{z}\}$, whose only nonzero structure constants satisfy $\mathbb{C}_{i,d+j}^{2d+1} = \delta_{ij}$ and $\mathbb{C}_{d+j,i}^{2d+1} = -\delta_{ij}$. From these relations one immediately sees that the derived algebra is $\mathfrak{h}^{(1)} = [\mathfrak{h}, \mathfrak{h}] = \mathbb{R}\mathfrak{z}$, and because $\mathfrak{z}$ is central we also have $\mathfrak{h}^{(2)} = [\mathfrak{h}, \mathfrak{h}^{(1)}] = 0$. Equivalently $\mathrm{ad}_X \circ \mathrm{ad}_Y \equiv 0$ for all $X, Y \in \mathfrak{H}(d)$. Therefore the Magnus expansion truncates exactly at second order, $\Omega^{(m)} \equiv 0 \ (m \geq 3)$, and this truncation is exact independently of the representation and of the numerical quadrature used.

To evaluate the surviving terms, write $\mathbb{A}(t) = \sum_{i=1}^d \alpha_{\mathfrak{q}_i}(t)\mathfrak{q}_i + \sum_{i=1}^d \alpha_{\mathfrak{p}_i}(t)\mathfrak{p}_i + \alpha_{\mathfrak{z}}(t)\mathfrak{z}$. The first Magnus term is its time integral

$$\Omega^{(1)} = \sum_{i=1}^d \Big( x_i \mathfrak{q}_i + y_i \mathfrak{p}_i \Big) + z^{(1)} \mathfrak{z}, \quad x_i := \int \alpha_{\mathfrak{q}_i}, \ y_i := \int \alpha_{\mathfrak{p}_i}, \ z^{(1)} := \int \alpha_{\mathfrak{z}}.$$

Since all commutators lie in the central direction, the second Magnus term contributes only to $\mathfrak{z}$:

$$\Omega^{(2)} = \tfrac{1}{2} \int_{t_k}^{t_{k+1}} \int_{t_k}^{\tau_1} [\mathbb{A}(\tau_1), \mathbb{A}(\tau_2)] \, d\tau_2 \, d\tau_1 = \tfrac{1}{2} \Big( \sum_{i=1}^d I_i \Big) \mathfrak{z}$$

where $I_i = \int_{t_k}^{t_{k+1}} \int_{t_k}^{\tau_1} \big( \alpha_{\mathfrak{q}_i}(\tau_1) \alpha_{\mathfrak{p}_i}(\tau_2) - \alpha_{\mathfrak{p}_i}(\tau_1) \alpha_{\mathfrak{q}_i}(\tau_2) \big) d\tau_2 \, d\tau_1$. Thus $(x_i, y_i)$ arise solely from $\Omega^{(1)}$, while the central coordinate receives the antisymmetric bilinear correction $\tfrac{1}{2} \sum_i I_i$.

We discretize the segment using quadrature nodes $\{\tau_\ell\}_{\ell=1}^Q$ with weights $w_\ell$ for single integrals and lower-triangular weights $W_{\ell m}$ for double integrals. Writing $x_i[\ell] \approx \alpha_{\mathfrak{q}_i}(\tau_\ell)$ and $y_i[\ell] \approx \alpha_{\mathfrak{p}_i}(\tau_\ell)$, we obtain

$$x_i = \sum_{\ell=1}^Q w_\ell x_i[\ell], \qquad y_i = \sum_{\ell=1}^Q w_\ell y_i[\ell], \qquad z^{(1)} = \sum_{\ell=1}^Q w_\ell \alpha_{\mathfrak{z}}[\ell].$$

Let $L \in \mathbb{R}^{Q \times Q}$ be the strictly lower-triangular matrix defined by $L_{\ell m} = W_{\ell m}$ for $m \leq \ell$ and 0 otherwise. Then

$$z^{(2)} = \tfrac{1}{2} \sum_{i=1}^d \Big( x_i^\top L y_i - y_i^\top L x_i \Big) = \tfrac{1}{2} \sum_{i=1}^d \sum_{\ell=1}^Q \sum_{m=1}^\ell W_{\ell m} \big( x_i[\ell] y_i[m] - y_i[\ell] x_i[m] \big).$$

Thus the entire second-order computation reduces to $d$ evaluations of the same bilinear form defined by $L$. The first-order cost is $O(Qd)$ because it consists of $B = 2d + 1$ dot products of length $Q$. The second-order term dominates: for a generic lower-triangular matrix with $(Q^2 + Q)/2$ nonzeros, each multiplication $Lv$ costs $\Theta(Q^2)$. Computing $x_i^\top(Ly_i) - y_i^\top(Lx_i)$ therefore requires $2\Theta(Q^2)$ operations per channel, plus $O(Q)$ for the final dot products, giving a total of $O(Q^2 d)$. This

matches the coefficient-mode Magnus bound $O(Q^2 \mathrm{nnz}(\mathbb{C}))$ since $\mathrm{nnz}(\mathbb{C}) = 2d$. The $O(Q^2 d)$ bound is optimal in the generic case. For a fixed channel $i$, the scalar $x_i^\top L y_i$ is a bilinear polynomial in $2Q$ variables with exactly $\frac{Q(Q+1)}{2}$ distinct monomials having generic coefficients. Any straight-line circuit computing this polynomial must account for all these monomials or their linearly independent combinations; thus $\Omega(Q^2)$ work is required. Since channels are independent, the total lower bound is $\Omega(Q^2 d)$.

For merging two leaf exponents $X = (x, y, z)$ and $Y = (\tilde{x}, \tilde{y}, \tilde{z})$, the identity $\mathrm{ad}^2 \equiv 0$ implies

$$\log(\exp(X)\exp(Y)) = \left(x + \tilde{x},\; y + \tilde{y},\; z + \tilde{z} + \tfrac{1}{2}\langle x, \tilde{y}\rangle - \tfrac{1}{2}\langle y, \tilde{x}\rangle\right),$$

so a merge requires $2d$ coordinate additions and one length-$d$ inner product; hence $O(d)$ operations. Touching all coordinates already requires $\Omega(d)$ operations, and the central update depends on $2d$ products, so $O(d)$ is optimal up to constants. Aggregating costs over all $M$ leaves, each leaf requires $\mathcal{C}_{\mathrm{leaf}}^{\mathrm{H}} = O(Qd) + O(Q^2 d)$, and a balanced binary scan performs $M - 1$ merges, each costing $O(d)$, yielding $\mathcal{W}_{\mathrm{H}} = O\big(Md(Q + Q^2 + 1)\big)$. The scan depth is $O(\log M)$. Storing all leaf exponents requires $O(Md)$ memory, while buffering only tree-level partial products reduces this to $O(d \log M)$. The stated bounds are therefore achieved and, by the arguments above, are optimal up to constant factors for generic quadrature weights. □

---

**Theorem B.4** (Unitary $\mathfrak{u}(d)$). *Let $B = d^2$ and assume matrix-mode evaluation with skew-Hermitian $\Omega^{[p]}$ at order $p \in \{2, 4\}$. Per segment, forming commutators and exponentials is dominated by matrix multiplications:*

$$\mathcal{C}_{\mathrm{leaf}}^{\mathrm{U}} = O(QD^3) + O(D^3), \quad \mathcal{C}_{\mathrm{merge}}^{\mathrm{U}} = O(D^3).$$

*Hence $\mathcal{W}_{\mathrm{U}} = O\big(M(Q + 2)D^3\big)$, $\mathcal{D}_{\mathrm{U}} = O(\log M)$, with memory $O(MD^2)$ (or $O(D^2 \log M)$ buffered). In* coefficient mode *one has the upper bound*

$$\mathcal{C}_{\mathrm{leaf}}^{\mathrm{U,coef}} = O(QB) + O\big(Q^2 \mathrm{nnz}(\mathbb{C})\big) + O\big(Q^3 B \Delta_{\mathbb{C}}^2\big),$$

*which is typically less favorable since $\mathrm{nnz}(\mathbb{C}) = \Theta(B^2)$ for dense $\mathfrak{su}(d)$ bases.*

---

*Proof.* For a single segment, the truncated Magnus exponent $\Omega^{[p]}$ consists of finitely many time-ordered integrals whose integrands are commutators of $A(t) \in \mathfrak{u}(d)$. Each commutator $[X, Y] = XY - YX$ requires two dense matrix multiplications and therefore costs $O(D^3)$ operations. Using a $Q$-point quadrature rule, the adopted leaf construction evaluates $O(Q)$ such commutator or matrix-multiplication terms per segment. In addition, computing the matrix exponential $\exp(\Omega^{[p]})$ is dominated by dense matrix multiplications and costs $O(D^3)$, hence $\mathcal{C}_{\mathrm{leaf}}^{\mathrm{U}} = O(QD^3) + O(D^3)$. In matrix mode, a scan merge corresponds to a single group product $U_2 U_1$, so in one dense matrix multiplication, giving $\mathcal{C}_{\mathrm{merge}}^{\mathrm{U}} = O(D^3)$. Substituting into Proposition B.2 yields

$$\mathcal{W}_{\mathrm{U}} = O\big(M(Q + 2)D^3\big), \qquad \mathcal{D}_{\mathrm{U}} = O(\log M).$$

If the Magnus expansion is evaluated in coefficient mode with basis size $B = d^2$, Proposition B.1 gives

$$\mathcal{C}_{\mathrm{leaf}}^{\mathrm{U,coef}} = O(QB) + O(Q^2 \mathrm{nnz}(\mathbb{C})) + O(Q^3 B \Delta_{\mathbb{C}}^2).$$

For dense bases of $\mathfrak{su}(d)$, $\mathrm{nnz}(\mathbb{C}) = \Theta(B^2)$. □

---

The cost is therefore dominated by matrix multiplications, while unitarity guarantees algebra-preserving updates without increasing scan depth. We state the resulting bounds directly, as the analysis follows standard matrix-commutator and GEMM-dominated arguments.

**Theorem B.5** (Projective/Quotient $\mathfrak{sl}(D) \subset \mathfrak{gl}(D)$). *Decompose $A = \alpha I + S$ with $S \in \mathfrak{sl}(D)$ and scalar channel $\alpha \in \mathbb{R}$. Then Magnus and scan decouple as a scalar prefix-sum for the center and an $SL(D)$ prefix-product for the traceless part. Consequently,*

$$\mathcal{C}^{\mathrm{SL}}_{\mathrm{leaf}} = O(QD^3) + O(D^3), \qquad \mathcal{C}^{\mathrm{SL}}_{\mathrm{merge}} = O(D^3),$$

*while the center adds only $O(Q) + O(1)$ per segment and merge. Thus*

$$\mathcal{W}_{\mathrm{SL}} = O\big(M(Q+2)D^3\big) + O(M), \qquad \mathcal{D}_{\mathrm{SL}} = O(\log M).$$

*Moreover, removing the center improves numerical conditioning; if the stable step-size increases by a factor $\gamma > 1$ under the traceless constraint, the total work scales as $\mathcal{W}_{\mathrm{SL}}/\gamma$ for a fixed horizon.*

*Proof.* Let the generator to be decomposed as $A(t) = \alpha(t)I + S(t)$, $S(t) \in \mathfrak{sl}(D)$, where $I$ spans the one-dimensional center. Since $[I, \cdot] = 0$, all structure constants involving the central direction vanish. By linearity of the Magnus expansion and centrality of $I$, the truncated Magnus exponent splits exactly as

$$\Omega^{[p]}(t_{k+1}, t_k) = \left(\int_{t_k}^{t_{k+1}} \alpha(\tau)\, d\tau\right) I + S^{[p]}(t_{k+1}, t_k), \qquad S^{[p]} \in \mathfrak{sl}(D).$$

All commutator terms depend only on $S(t)$ and therefore remain in $\mathfrak{sl}(D)$. Forming $S^{[p]}$ and evaluating $\exp(S^{[p]})$ are dominated by dense matrix multiplications, giving a cost $O(QD^3) + O(D^3)$ per segment. The central scalar requires only quadrature and costs $O(Q)$. For two consecutive segments with exponents $\Omega_i = \beta_i I + S_i$, one has

$$\exp(\Omega_2)\exp(\Omega_1) = \exp((\beta_2 + \beta_1)I)\exp(S_2)\exp(S_1),$$

so the central components combine by scalar addition, while the traceless components combine by matrix multiplication in $SL(D)$. Thus the merge cost is $O(D^3)$, with an additional $O(1)$ scalar update. Substituting into Proposition B.2 yields

$$\mathcal{W}_{\mathrm{SL}} = O\big(M(Q+2)D^3\big) + O(M), \qquad \mathcal{D}_{\mathrm{SL}} = O(\log M).$$

The linear scalar term does not affect the asymptotic complexity. Memory bounds coincide with the matrix-mode case. $\square$

For step-2 nilpotent algebras such as the Heisenberg algebra, all higher-order Magnus and BCH terms vanish, reducing both leaf construction and group merges to bilinear operations with linear complexity in the basis size, whereas for unitary and special linear algebras the commutator structure is preserved and the dominant cost is governed by matrix multiplications while the balanced scan retains logarithmic depth independently of the horizon, and in the projective setting separating the scalar center further decouples a prefix-sum channel from the traceless matrix dynamics, which can improve numerical stability in practice without changing asymptotic merge costs.

## C. Experimental Details

**Data generation.** *Met-Enkephalin.* Using OpenMM with the Amber14SB all-atom force field and explicit TIP3P-FB water, molecular dynamics simulations are performed for Met-Enkephalin starting from the experimentally resolved structure in PDB 1PLX RCSB Protein Data Bank (2004). Initial velocities are sampled from the Maxwell–Boltzmann distribution at a temperature of 300 K, and the system is propagated with a Langevin integrator using a time step of 2 fs. Throughout the simulation, atomic coordinates are recorded at intervals of 10 fs in order to construct a continuous trajectory suitable for long-horizon analysis. Under this protocol, molecular dynamics trajectories are generated, and the dataset used in all experiments is obtained by extracting the 1000 frames from the resulting trajectories so that preprocessing remains identical across all evaluation settings. *Alkane molecules.* Each system is initialized from a single PDB structure downloaded from the OPLS–molecules repository, and the molecular topology is constructed by converting the PDB structure into an RDKit molecular graph and creating an OpenMM `Topology` containing a single residue, with OpenFF 2.0.0 force field providing all non-periodic molecular mechanics parameters through the SMIRNOFF scheme. Simulations use a Langevin integrator with a 2 fs step size and a friction coefficient of $1$ ps$^{-1}$ and for every trajectory the positions are reset to the reference geometry while the initial velocities are sampled from a Maxwell–Boltzmann distribution at $T = 300/16$ K.

The detailed underlying hamiltonian dynamics are defined as the following:

$$H = \frac{1}{2} \sum_{i=1}^{N} \frac{\|\mathbf{p}_i\|^2}{m_i} + \sum_{\tau, n \in N_\tau} V_{\tau,n} \left[ 1 + \cos(n\varphi_\tau - \delta_{\tau,n}) \right]$$
$$+ \sum_{(i,j,k) \in \mathfrak{A}} \frac{1}{2} \kappa_{ijk} (\theta_{ijk} - \theta_{ijk}^{\text{eq}})^2 + \sum_{(i,j) \in \mathfrak{B}} \frac{1}{2} \nu_{ij} (r_{ij} - r_{ij}^{\text{eq}})^2. \tag{17}$$

**Dataset construction.** The dataset comprises trajectories for Met-Enkephalin, while 100 molecular trajectories are stored for alkane molecules as position–momentum sequences of shape $(S, M, 6N)$, where $S$ is the number of samples, $M = 1000$ is the rollout length and $N$ is the number of atoms. No center-of-mass alignment is applied because the systems contain a small number of atoms and the global orientation does not affect the learning task. All coordinate and momentum channels are normalized using statistics computed from the training portion of the dataset, which is split into training and validation sets using an 80:20 ratio after separating test trajectories for final evaluation.

**Model architecture.** Given a state $x_t \in \mathbb{R}^d$, a two layer MLP encoder maps $x_t$ to a latent vector $z_t \in \mathbb{R}^K$, and a decoder with the same depth and width maps latent states back to the original space. Here, $d$ denotes the dimension of the system state, whereas $K$ denotes the dimension of the latent state on which the Lie-algebraic evolution is performed. In all reported experiments, we set $K = d$. Time is discretized into $M$ uniform steps on $[0, T]$ with step size $h = T/M$, and at each step $t_m$ the model predicts a Lie algebra element $A_\theta(t_m) = \sum_{a=1}^{|\mathfrak{h}|} \alpha_a(t_m) \mathfrak{h}_a$, where the coefficients $\alpha(t_m) \in \mathbb{R}^{|\mathfrak{h}|}$ are produced by a sinusoidal time embedding with $n_{\text{freq}}$ frequencies followed by a two layer MLP. The basis $\{\mathfrak{h}_a\}_{a=1}^{|\mathfrak{h}|}$ spans a fixed Lie subalgebra $\mathfrak{h} \subseteq \mathfrak{gl}(K)$ and determines the structure constants $C_{ab}^c$ used in the Baker–Campbell–Hausdorff composition. For a sequence of generators $\{A_\theta(t_m)\}_{m=0}^{M-1}$ we form segment level increments $\Delta U_m = h A_\theta(t_m)$ and approximate their composition with a second order Baker–Campbell–Hausdorff expansion, and one step Koopman propagator is evaluated by a $6^{th}$ order truncated matrix exponential. Cumulative propagators are obtained by applying the associative prefix scan to the Lie algebraic increments, and the scan procedure runs in depth $O(\log_2 M)$. We explore three Lie algebra choices, with all model variants share the same encoder, decoder, time embedding and optimizer, and differ only in the choice of Lie subalgebra and the resulting structure constants. The projective variant constructs a learned basis in $\mathfrak{gl}(K)$ and projects every generator onto its traceless component so that all updates lie in $\mathfrak{sl}(K)$. The unitary variant builds an orthonormal basis of skew Hermitian matrices that represents $\mathfrak{u}(K)$ and discards the central direction corresponding to the identity. The Heisenberg variant assumes that the number of generators satisfies $|\mathfrak{h}| = 2d + 1$ and instantiates the analytic Heisenberg algebra with basis $\{q_1, \dots, q_d, p_1, \dots, p_d, z\}$. In this case the only nonzero structure constants are $C_{i,d+i,z} = 1$ and $C_{d+i,i,z} = -1$ for $1 \le i \le d$, and the nilpotent property ensures that the Baker–Campbell–Hausdorff series truncates exactly at second order.

**Training setup.** All experiments are run on a single RTX 5000 Ada GPU using Python 3.10.18 with PyTorch 2.5.1 and CUDA 12.4, and CUDA synchronization is enabled during evaluation to obtain reliable timing measurements. We train all models with the Adam optimizer using a learning rate of $10^{-3}$ and the default momentum coefficients, without applying any learning rate decay, and gradients are clipped to a unit $\ell_2$ norm. Training proceeds on fixed windows of length $M$, and dataset provides windows taken from the first $M$ frames of its normalized trajectories. Model checkpoints are chosen by monitoring the forward-trajectory mean squared error, which is evaluated once at every iteration.

**Baseline implementations.** All baseline models follow the official public implementations of the respective authors. Each model is evaluated on the same preprocessed data used for our method, with every trajectory flattened and normalized with the statistics described in the data preprocessing section.

- **Koopa**: https://github.com/thuml/Koopa
- **KoVAE**: https://github.com/azencot-group/KoVAE
- **KooNPro**: https://github.com/Rrh-Zheng/Koonpro
- **KoNODE**: https://github.com/Baitie00/KoNODE
- **MamKO**: https://openreview.net/forum?id=hNjCVVm0EQ
- **SEGNO**: https://github.com/yliukj/SEGNO
- **EqMotion**: https://github.com/MediaBrain-SJTU/EqMotion

- **GeoTDM**: https://github.com/hanjq17/GeoTDM

We use concatenated position–momentum vectors for models that require momentum, and both channels follow the same normalization protocol with the rollout length fixed to $M = 1000$ in all settings. We employed the published architectural components without modification for Koopman-based baselines (Koopa, KoVAE, KooNPro, KoNODE, MamKO). Input dimensionality and horizon-related parameters are adjusted only to ensure compatibility with the unified preprocessing pipeline, and no additional regularization terms are introduced. We modify only scalar optimization parameters whenever a baseline fails due to numerical instability, such as the learning rate or the number of training iterations. These revisions do not change any structural aspect of the original algorithm. The runtime of models with heavy temporal complexity, such as attention-based or latent-flow methods, reflects their original computational behavior. We followed the same preprocessing steps for generative and probabilistic sequential baselines (EqMotion, GeoTDM), with Equivariant layers, diffusion processes, and solver configurations remain identical to the official implementations.

## D. Additional Experimental Results

**RMSD Analysis of Alkane Trajectories.** Beyond pointwise prediction error, long-horizon molecular simulations require preservation of geometric consistency to remain physically meaningful. To assess whether predicted trajectories stay within a physically plausible regime over extended rollouts, pairwise root mean squared deviation between predicted and reference atom coordinates is evaluated over time. This analysis complements accuracy comparisons by explicitly quantifying structural drift accumulated as the rollout horizon increases. Figure 6 reports the temporal evolution of pairwise RMSD over the entire horizon rollouts for alkane trajectories.

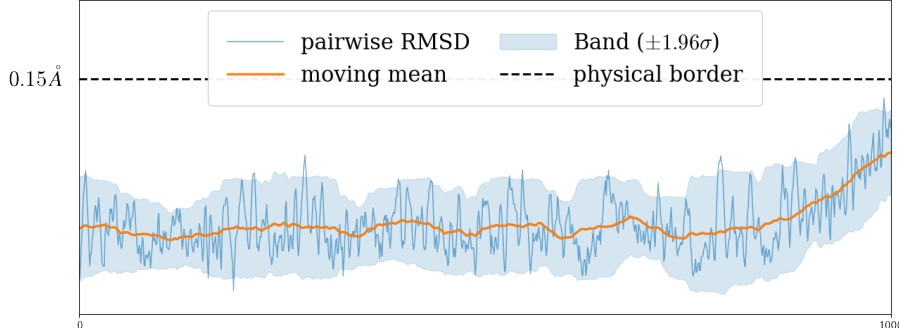

*Figure 6.* Backbone pairwise RMSD between predicted and ground-truth trajectories with local variability band.

The solid curve denotes a moving average of RMSD, while the shaded region represents the corresponding uncertainty band defined by $\pm 1.96\sigma$. Across the entire rollout, the mean RMSD remains well below the commonly adopted physical tolerance threshold of 0.15 Å indicated by the dashed reference line. Although a gradual increase in RMSD is observed toward the end of the rollout, the associated uncertainty band largely stays within the physically admissible range. The absence of abrupt divergence in RMSD further suggests that error accumulation remains controlled rather than exhibiting unstable growth. These observations indicate that the proposed model preserves geometric fidelity over long horizons and avoids unphysical structural drift even when predictions extend substantially beyond the training window.

**Lotka–Volterra System.** The Lotka–Volterra system constitutes a classical nonlinear ordinary differential equation model that was originally introduced to describe predator–prey interactions in population dynamics. In its two-dimensional form, the system is governed by $\dot{x}(t) = \alpha x(t) - \beta x(t)y(t),\ \dot{y}(t) = \delta x(t)y(t) - \gamma y(t)$, where $x(t)$ and $y(t)$ denote the prey and predator populations, respectively, and the positive coefficients $\alpha, \beta, \gamma, \delta$ control intrinsic growth and interaction rates. Although the governing equations appear simple, the resulting trajectories exhibit nonlinear oscillatory behavior and long-range temporal dependencies that pose nontrivial challenges for sequence modeling and extrapolation.

To evaluate the ability of the proposed models to reconstruct and extrapolate nonlinear dynamics over extended horizons, trajectories are generated using fixed coefficients $\alpha = 1.1$, $\beta = 0.4$, $\delta = 0.1$, and $\gamma = 0.4$. Numerical integration is performed on the interval $[0, 10]$ using a classical fourth-order Runge–Kutta scheme with $M = 1000$ uniform time steps. Initial states are sampled independently from the square $[0.3, 1.8]^2$, and trajectories are generated on the fly during training without employing an explicit dataset split. This setup allows direct evaluation of long-horizon prediction accuracy while avoiding confounding effects introduced by finite training datasets.

*Table 3.* **Performance comparison on the Lotka–Volterra system.** Inference runtime, training time, best mean squared error, and $\Delta$Deg are reported under rollout lengths $M = 100$ and $M = 1000$.

| Method | Inference Time (s) | | Train Time (s) | | Best MSE $\downarrow$ | | $\Delta$Deg |
|---|---|---|---|---|---|---|---|
| | $M = 100$ | $M = 1000$ | $M = 100$ | $M = 1000$ | $M = 100$ | $M = 1000$ | |
| *(A) Koopman based models (*Low* Accuracy,* High* Efficiency)* | | | | | | | |
| Koopa (Liu et al., 2023) | $1.56\times10^{-2}$ | $1.78\times10^{-2}$ | $1.17\times10^{-1}$ | $1.40\times10^{-1}$ | $2.65\times10^{-2}$ | $7.81\times10^{0}$ | $3.53\times10^{2}$ |
| KoVAE (Naiman et al., 2024) | $4.40\times10^{-2}$ | $2.44\times10^{-1}$ | $1.58\times10^{-1}$ | $9.09\times10^{-1}$ | $1.97\times10^{-2}$ | $2.57\times10^{-2}$ | $7.51\times10^{0}$ |
| KooNPro (Zheng et al., 2025) | $3.22\times10^{-1}$ | $2.06\times10^{0}$ | $7.02\times10^{-1}$ | $6.05\times10^{0}$ | $6.42\times10^{-1}$ | $5.51\times10^{-1}$ | $7.40\times10^{0}$ |
| KoNODE (Bai & Ding, 2025) | $1.80\times10^{-1}$ | $1.51\times10^{0}$ | $1.05\times10^{0}$ | $8.05\times10^{0}$ | $5.37\times10^{-2}$ | $3.42\times10^{-2}$ | $4.88\times10^{1}$ |
| MamKO (LI et al., 2025) | $6.43\times10^{-1}$ | – | $1.11\times10^{-1}$ | – | $2.02\times10^{-2}$ | – | – |
| *(B) Generative model based probabilistic sequential models (*High* Accuracy,* Low* Efficiency)* | | | | | | | |
| EqMotion (Xu et al., 2023) | $9.86\times10^{-2}$ | $6.97\times10^{0}$ | $1.21\times10^{-1}$ | $6.95\times10^{0}$ | $3.86\times10^{-2}$ | $8.48\times10^{-2}$ | $1.26\times10^{1}$ |
| GeoTDM (Han et al., 2024) | $8.52\times10^{1}$ | $8.83\times10^{2}$ | $3.06\times10^{-2}$ | $1.07\times10^{-1}$ | $7.67\times10^{-3}$ | $1.83\times10^{-2}$ | $8.34\times10^{0}$ |
| Ours (Heisenberg) $\mathfrak{H}(d)$ | $\mathbf{2.91\times10^{-3}}$ | $\mathbf{1.67\times10^{-2}}$ | $\mathbf{7.13\times10^{-3}}$ | $\mathbf{3.91\times10^{-2}}$ | $8.23\times10^{-3}$ | $9.27\times10^{-3}$ | $6.18\times10^{0}$ |
| Ours (Unitary) $\mathfrak{u}(d)$ | $5.06\times10^{-3}$ | $1.80\times10^{-2}$ | $1.10\times10^{-2}$ | $4.21\times10^{-2}$ | $\mathbf{1.84\times10^{-3}}$ | $\mathbf{1.35\times10^{-3}}$ | $\mathbf{2.80\times10^{0}}$ |
| Ours (Projective) $\mathfrak{sl}(K)$ | $3.06\times10^{-3}$ | $1.85\times10^{-2}$ | $7.48\times10^{-3}$ | $4.30\times10^{-2}$ | $4.99\times10^{-3}$ | $1.00\times10^{-2}$ | $1.15\times10^{1}$ |

To characterize the impact of increasing temporal horizons on both computational efficiency and predictive accuracy, performance is evaluated under two rollout lengths $m_0 = 100$ and $m_1 = 1000$. Degradation induced by horizon extension is quantified using $\Delta\text{Deg} := \frac{\tau(m_1)}{\tau(m_0)} \cdot \frac{\mathcal{E}_\star(m_1)}{\mathcal{E}_\star(m_0)}$, where $\tau(m)$ denotes the wall-clock training time per iteration normalized by the rollout length $m$, and $\mathcal{E}_\star(m)$ denotes the minimum mean squared error achieved under horizon $m$. This metric jointly reflects the growth of computational cost and the accumulation of long-term prediction error as the horizon increases.

Table 3 reports inference time, training time, best achieved mean squared error, and the corresponding degradation metric for all compared methods. As shown in the table, the proposed models consistently achieve superior accuracy–efficiency trade-offs under long-horizon evaluation. Among the three Lie algebra variants, the unitary model attains the lowest prediction error at both horizons while maintaining substantially lower computational cost than prior approaches. In contrast, several Koopman-based baselines exhibit pronounced performance degradation as the rollout length increases, despite favorable efficiency at shorter horizons.

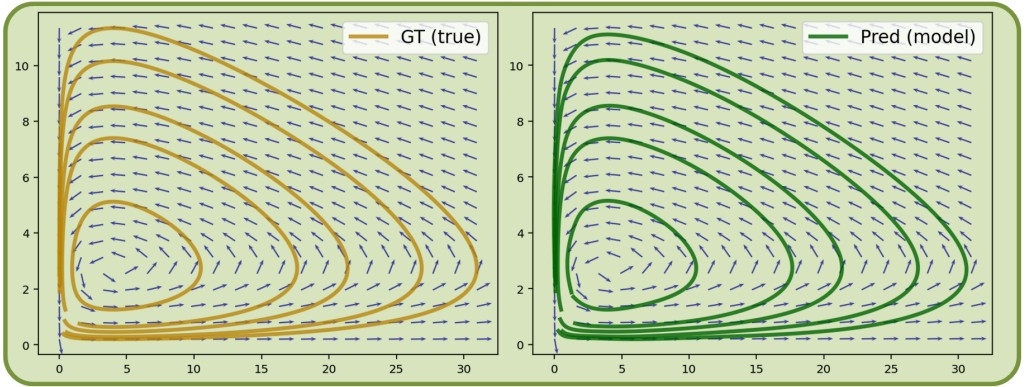

*Figure 7.* **Qualitative comparison on the Lotka–Volterra system.**

Qualitative results further illustrate the stability of the proposed approach under extended rollouts. As shown in Figure 7 that visualizes five randomly sampled trajectories together with the corresponding model predictions, the proposed models preserve both oscillatory structure and phase alignment with the ground truth trajectories. These results indicate that Lie-algebraic composition combined with associative prefix scan enables robust long-term propagation without incurring prohibitive computational overhead.

**Evaluation on ETT.** We further evaluate our framework on the ETT benchmark (Zhou et al., 2021) to assess its applicability beyond molecular dynamics. As shown in Table 4, all three variants achieve lower Best MSE and shorter inference time than all baselines. The unitary variant records the lowest Best MSE of $6.58\times10^{-1}$, while the Heisenberg variant achieves the shortest inference time of $6.04\times10^{-2}$ seconds, providing an $8.3\times$ speedup over the fastest baseline. These results demonstrate a consistent accuracy–efficiency advantage on the ETT forecasting task.

*Table 4.* **Performance comparison on the ETT benchmark.**

| Method | Best MSE ↓ | Inference Time (s) ↓ |
|---|---|---|
| Koopa (Liu et al., 2023) | $6.82 \times 10^{-1}$ | $5.00 \times 10^{-1}$ |
| KoVAE (Naiman et al., 2024) | $1.28 \times 10^{0}$ | $1.44 \times 10^{0}$ |
| KooNPro (Zheng et al., 2025) | $8.90 \times 10^{-1}$ | $1.31 \times 10^{0}$ |
| KoNODE (Bai & Ding, 2025) | $6.77 \times 10^{-1}$ | $3.09 \times 10^{0}$ |
| MamKO (LI et al., 2025) | $8.67 \times 10^{-1}$ | $1.97 \times 10^{0}$ |
| EqMotion (Xu et al., 2023) | $8.14 \times 10^{-1}$ | $1.09 \times 10^{1}$ |
| SEGNO (Liu et al., 2024) | $1.86 \times 10^{0}$ | $6.08 \times 10^{0}$ |
| GeoTDM (Han et al., 2024) | $1.69 \times 10^{0}$ | $7.43 \times 10^{1}$ |
| Ours (Heisenberg) $\mathfrak{h}(d)$ | $6.59 \times 10^{-1}$ | $\mathbf{6.04 \times 10^{-2}}$ |
| Ours (Unitary) $\mathfrak{u}(d)$ | $\mathbf{6.58 \times 10^{-1}}$ | $6.59 \times 10^{-2}$ |
| Ours (Projective) $\mathfrak{sl}(K)$ | $6.59 \times 10^{-1}$ | $6.16 \times 10^{-2}$ |

# E. Algorithmic Details

---

**Algorithm 1** Parallel Prefix Scan on Lie Group

---

**Require:** Segment-wise Lie-algebra elements $\{\Omega_m\}_{m=0}^{M-1}$, group exponential $\mathrm{Exp} : \mathfrak{g} \to G$, group logarithm $\mathrm{Log} : G \to \mathfrak{g}$, associative group product $\otimes$ on $G$, identity element $I \in G$

**Ensure:** Cumulative generators $\{\Theta_m\}_{m=0}^{M}$

  {Pad to next power of two}

  $M' \leftarrow 2^{\lceil \log_2 M \rceil}$

  **for** $m = M$ **to** $M' - 1$ **do**

    $\Omega_m \leftarrow 0$

  **end for**

  {Initialize leaves in the Lie group}

  **for** $m = 0$ **to** $M' - 1$ **do**

    $G_m^{(0)} \leftarrow \mathrm{Exp}(\Omega_m)$

  **end for**

  $L \leftarrow \log_2 M'$

  {Upsweep: compute subtree products in $G$}

  **for** $\ell = 0$ **to** $L - 1$ **do**

    **for** $j = 0$ **to** $M'/2^{\ell+1} - 1$ **do**

      $G_j^{(\ell+1)} \leftarrow G_{2j}^{(\ell)} \otimes G_{2j+1}^{(\ell)}$

    **end for**

  **end for**

  {Downsweep: exclusive prefix scan in $G$}

  $P_0^{(L)} \leftarrow I$

  **for** $\ell = L - 1$ **downto** $0$ **do**

    **for** $j = 0$ **to** $M'/2^{\ell+1} - 1$ **do**

      $P_{2j}^{(\ell)} \leftarrow P_j^{(\ell+1)}$

      $P_{2j+1}^{(\ell)} \leftarrow P_j^{(\ell+1)} \otimes G_{2j}^{(\ell)}$

    **end for**

  **end for**

  {Extract prefixes and map back to the Lie algebra}

  **for** $m = 0$ **to** $M$ **do**

    $U_m \leftarrow P_m^{(0)}$

    $\Theta_m \leftarrow \mathrm{Log}(U_m)$

  **end for**

  **return** $\{\Theta_m\}_{m=0}^{M}$

---

We describe the parallel prefix-scan algorithm used to accumulate segment-wise Lie-algebra elements into cumulative generators, as introduced in Section 3. The algorithm implements the associative composition and enables logarithmic-depth evaluation over long temporal horizons. Given a partition $s = t_0 < t_1 < \cdots < t_M = t$, the model produces a sequence of Lie-algebra elements $\{\Omega_m\}_{m=0}^{M-1}$, $\Omega_m \approx \Omega(t_{m+1}, t_m)$, represented in coefficient form with respect to a fixed basis of the chosen Lie subalgebra. The goal of the scan is to compute cumulative generators $\Theta_0 := 0$, $\Theta_m = \Omega_{m-1} \oplus \Omega_{m-2} \oplus \cdots \oplus \Omega_0$, $m = 1, \ldots, M$, where $\oplus$ denotes truncated algebraic composition. The scan is evaluated using a balanced binary tree.

