# OpenReview forum: "Lie-Algebraic Acceleration of Neural Koopman Dynamics"
_ICML.cc/2026/Conference — ICML 2026 regular_

### Official Review · Reviewer_5kvc · 2026-03-03

**Soundness:** 3
**Presentation:** 3
**Significance:** 3
**Originality:** 3
**Overall Recommendation:** 4
**Confidence:** 3

**Summary:**

In this paper, a novel Lie-algebraic method is designed to parameterize the Koopman operator of dynamical systems. By constraining the generator $\mathcal{L}$ to finite-dimensional Lie subalgebras, i.e., Heisenberg Algebra, Unitary Algebra, and Projective Algebra, the authors reformulate the nested commutators of the Magnus expansion into scalar operations via structure constants, which makes the computation tractable. Furthermore, by leveraging two essential algebraic properties, i.e., transitivity and associativity, the proposed method can conduct long-term prediction in a parallel manner, which significantly reduces the inference time and cumulative error.

**Compliance With Llm Reviewing Policy:**

Affirmed.

**Final Justification:**

Based on the authors' response, my concern has been addressed. I recommend the author further clarify the dataset setting in the revised paper for avoiding potential confusion.

**Key Questions For Authors:**

1. In Definition 2.1, the independent variable for $\alpha_a$ is $z_0$, whereas in Proposition 2.2, the independent variable for $\alpha_c$ is $\mathbf{x_0}$. Is this an intentional distinction, or is it a typo?

2. The authors utilize the $m$-th order contribution of the Magnus expansion. How is the exact value of $m$ determined in practice?

3. In Equation (7), should the variable $\mathbf{x}$ be replaced with $\mathbf{z}$?

4. According to the main manuscript, only the $\alpha$ is parameterized by a NN. However, the ''Model architecture'' part in Appendix C reveals that the encoder and decoder (mapping between $\mathbf{x}$ and $\mathbf{z}$) are also parameterized by NNs. Why is there such a discrepancy between the main manuscript and the appendix?

5. Although the proposed method demonstrates superior results in Table 1, the right part of Figure 3 illustrates an obvious deviation between the model prediction and the GT. Is this significant deviation reasonable or expected under the experimental settings?

6. The dataset is stated to be partitioned into a training set and a validation set. Could the authors clarify the exact protocol for utilizing these two datasets during training and evaluation? Specifically, how is the optimal model selected?

7. What general principles should be followed when selecting an appropriate Lie subalgebra for a dynamical system?

**Limitations:**

It may be helpful to discuss the limitation of the proposed method in generalizing across various dynamical systems due to its reliance on pre-defined Lie subalgebras.

**Strengths And Weaknesses:**

**Strengths:**
1. The use of a Lie-algebraic framework to parameterize the Koopman operator is novel and theoretically elegant.
2. Leveraging Lie-algebraic properties, the proposed method utilizes the prefix-scan algorithm to accelerate inference and reduce cumulative prediction errors.
3. The study explores three finite-dimensional Lie subalgebras applicable to the proposed method.

**Weaknesses:**
1. The proposed method is difficult to generalize across various dynamical systems. This is because it relies on the pre-defined Lie subalgebras, while selecting an appropriate subalgebra requires a comprehensive understanding of the modeled system. Furthermore, even if an appropriate Lie subalgebra can be chosen empirically, deriving its corresponding structure constants is still challenging.

2. While the theoretical derivations in the paper are rigorous, numerous simplifications are introduced in the practical implementation. For instance, a $6^{th}$ order truncated matrix exponential is used. These simplifications may render some of the good theoretical properties invalid in practical applications.

3. There are some limitations in the paper presentation. Some symbols, such as $z$, are introduced without prior definition. Additionally, certain evaluation metrics are not adequately explained. For example, the specific meaning of "Best" in the "Best MSE" metric reported in the tables is not detailed. Moreover, the design of the tables in the paper is too complex to understand. Finally, the main text of the paper focuses almost exclusively on theoretical discussions, without providing a detailed description of the actual method proposed based on these theories, which is not mentioned until the 'Model architecture' part of Appendix C.

---

> ### Author Rebuttal · Authors · 2026-03-30
>
> We sincerely thank you for the thoughtful and constructive feedback. We appreciate the opportunity to clarify the concerns raised and to further improve the presentation of our work. Below, we respond to each point directly, provide additional explanation where the current exposition may have been unclear, and describe the revisions we will make in the final version. We hope that our responses sufficiently address your concerns and help improve the clarity and overall quality of the final version.
>
> **Q. The proposed method is difficult to generalize…What general principles should be followed…It may be helpful to discuss the limitation…**
>
> We agree that selecting a system-specific Lie subalgebra for an arbitrary dynamical system is an important problem. This paper instead focuses on canonical Lie subalgebras and examines how their algebraic and computational properties influence learning efficiency, stability, and long-horizon rollout performance. In our setting, the structure constants are not additional empirical quantities that must be inferred from data. For the canonical algebras considered here, they are known analytically and directly determine the corresponding Magnus and scan constructions.
>
> **Q. Numerous simplifications are introduced… How is the exact value of determined in practice?**
>
> Our practical implementation uses standard numerical approximations for efficiency, but they affect only the numerical realization, not the underlying structure of the method. In particular, we use a 6th-order truncated matrix exponential, as we found it to provide good practical results, with no meaningful improvement when increasing the order. For Magnus expansion, we use a second-order Magnus truncation in all experiments. This is because the first omitted term is only third order in $\Delta t$, which is negligible under fine temporal discretization. For the Heisenberg algebra, this truncation is even exact due to step-2 nilpotency.
>
> **Q. certain evaluation metrics are not adequately explained... clarify the exact protocol for utilizing… how is the optimal model selected?**
>
> We use the training set for optimization and the validation set for model selection. We evaluated all methods using the checkpoint selected by the lowest validation MSE. In the tables, “Best MSE” denotes the average frame-wise MSE over the full trajectory generated by the model. We will clarify this in the final manuscript.
>
> **Q. Some symbols are introduced without…The design of the tables in the paper is too complex…**
>
> Thank you for this helpful comment. We will explicitly define the main variables, including $z$, when they first appear, make the tables easier to understand by reducing the number of colors and adding additional explanations about the metrics in the main text.
>
> **Q. In Definition 2.1, the independent variable for $\alpha_a$ is $z_0$…**
>
> Thank you for pointing this out. The use of $z_0$ in Definition 2.1 and $x_0$ in Proposition 2.2 was intended to distinguish the latent variable from the input state, but this distinction is not sufficiently explained. We will clarify this relation in the final version.
>
> **Q. In Equation (7), should the variable $\mathbf x$…**
>
> Equation (7) is correct as written with $x$. It defines the trajectory prediction error in the data space, and both the Problem Statement and Definition 2.1 describe the state evolution in terms of $x(t)$.
>
> **Q. the main text of the paper focuses almost…Why is there such a discrepancy between…**
>
> We appreciate your observation. The emphasis in the main text was on the Lie-algebraic parameterization of the generator coefficients $\alpha$, which made the practical implementation appear narrower than it actually is. In the implemented model, the encoder and decoder between $x$ and $z$ are also neural components, as described in Appendix C. We agree that this organization can create confusion between the theoretical formulation and the full model architecture. To address this, we will move the key contents of the current “Model architecture” part of Appendix C into a new subsection of Section 3, where we will explicitly summarize the full implemented pipeline.
>
> **Q. The right part of Figure 3 illustrates an obvious deviation…**
>
> While the right part of Figure 3 shows visible deviation from the GT, this does not contradict the results in Table 1. Since Ramachandran maps visualize torsion angles, relatively small local differences can appear visually amplified.

---

> > ### Author Rebuttal · Reviewer_5kvc · 2026-04-02
> >
> > Thanks for your response. My concerns have been mostly addressed, except for one. Based on your response, the model is optimized on the training set and selected on the validation set. Meanwhile, as discussed in the "Dataset construction" part in the Appendix C, the dataset is only split into a training set and a validation set. Therefore, it is confusing which dataset the selected model is tested on.

---

> > > ### Author Response · Authors · 2026-04-05
> > >
> > > Thank you for your thoughtful follow-up and continued engagement. We apologize for not clearly describing the dataset split and evaluation protocol in our earlier response and in Appendix C. The reported 80:20 split in Appendix C does not refer to the final partition of the entire dataset. Rather, it refers to an internal split used to define the training and validation sets within the training portion, after setting aside a separate test set. Both the training set and the test set were constructed following the same data generation protocol in Appendix C. In our experimental pipeline, the model was trained exclusively on the training set, the validation set was used for model selection, and the final reported metrics were computed on the test set. We agree that this was not stated clearly enough in the current version, and we will revise the manuscript to describe this protocol more clearly.

---

### Official Review · Reviewer_Yhua · 2026-03-10

**Soundness:** 3
**Presentation:** 3
**Significance:** 2
**Originality:** 2
**Overall Recommendation:** 4
**Confidence:** 3

**Summary:**

This paper develops a Lie-algebraic approach for approximating the Koopman operator of non-autonomous dynamical systems. This enables a rigorous approximation, but is computationally expensive. The authors utilize a prefix-scan to accelerate their approximation. The authors demonstrate the utility of their approach on 2 molecular dynamics benchmark data sets.

**Compliance With Llm Reviewing Policy:**

Affirmed.

**Final Justification:**

The authors addressed my concerns and the added discussion and numerical results make me confident to recommend accepting this work.

**Key Questions For Authors:**

I think for me to fully understand the contribution of this work, I would like the following answered:

1. How does the theoretical aspects of this work compare to the previous literature on symmetries and Lie-algebras for Koopman (e.g., Giannakis's work - this is not me, I promise)?

2. Does the authors' approach work better than Koopa (and other methods) on the kind of benchmark time-series that were used to validate it in the original papers?

**Limitations:**

The authors were transparent about their Lie-algebra framework being computationally intensive (this motivated the development of the prefix scan). But I think there could be more discussion on possible limitations of the method (how might it struggle with very noisy data or data that exhibits strong dynamical regime changes?). In addition, discussing limitations on the type of data studied would be good.

**Strengths And Weaknesses:**

## Strengths:

1. This work is a theoretically grounded approach for studying non-autonomous dynamical systems, which - in general - is an area less explored in the Koopman community.

2. The authors develop a computational framework that enables them to use their mathematical machinery on real-world problems in an efficient way.

3. The authors find strong results relative to the models they compare against.

4. The authors have nice figures that help explain the framework to the readers.

## Weaknesses:

1. The authors discuss a few recent machine learning works that leverage Koopman operator theory (e.g., Koopa, KoVAE), but there is very little mention of the large body of Koopman literature that has studied symmetries and algebras for Koopman (e.g., Salova et al. Chaos (2019), Giannakis Research in Mathematical Sciences (2021), Sechi et al. Multiscale Modeling and Simulation (2021), Freeman et al. PNAS (2023),  Vale et al. arXiv (2025) , Giannakis and Valva, Physica D (2025)). Given this work, I think that it is: 1) necessary the authors acknowledge that there has been efforts in leveraging Lie-algebras and other algebra-theoretic approaches to provide structure to improve the Koopman operator approximations; 2) discuss how their work fits into this past literature.

2. It was not clear to me why the authors chose the data sets they did to test their method on. Molecular dynamics have been studied by the Koopman community (which the authors should make more apparent/cite - Nuske and Klus Journal of Chemical Physics (2023), Kostic et al. NeurIPS (2022), Devergne et al. NeurIPS (2024),  Kostic et al. NeurIPS (2024), but Koopa and KoVAE were never tested on these. While I think this makes the authors findings interesting (i.e., there are domains where these state-of-the-art approaches are beat by the Lie-algebra approach), I think it would be appropriate and further revealing to study their approach on some of the same time-series that these methods were tested on.

3. I found Figure 3 hard to interpret, for several reasons. First, GT is not defined. I assume this is "ground-truth", but maybe not? Second, Ramachandran map was not explained. And third, There is no comparison with other models. How "badly" does a model that has poor accuracy look compared to GT?

## Minor points
1. $A$ is not explicitly defined in Def. 2.1
2. IVP is mentioned but not defined

---

> ### Author Rebuttal · Authors · 2026-03-30
>
> We appreciate your careful reading and thoughtful comments. Your review was helpful in clarifying where the manuscript should be strengthened, both in the presentation of the main ideas and in the empirical support for the paper. In response, we have focused our revision on improving the motivation, clarifying its scope, and adding further experimental results to better support the claims. Below, we describe these planned revisions in more detail. We hope that, if you find the responses and planned revisions satisfactory, this will be reflected in a more positive final evaluation.
>
> **Q. The authors discuss a few recent… How does the theoretical aspects…**
>
> The cited literature is related in spirit, but its technical objective is different from ours. For example, Salova et al. study how known symmetries of the underlying dynamics are reflected in Koopman operators and in EDMD/kernel-DMD approximations. Their main payoff is a representation-theoretic block structure that improves symmetry-adapted spectral approximation, not a learned Lie-group propagator. Likewise, Giannakis’s work and the later Giannakis–Valva spectral approximation line focus on constructing coherent observables and consistent approximations of Koopman operators/generators through delay embeddings, kernel integral operators with emphasis on spectral recovery and operator consistency rather than neural propagation. By contrast, our contribution is to learn and parameterize the time-varying generator itself inside a finite-dimensional Lie subalgebra, construct segmentwise finite-time propagators that remain in the associated Lie group through Magnus-type exponentiation, and then exploit the associativity of Lie-group composition to obtain a prefix-scan architecture for temporally parallel long-horizon propagation. Thus, the core concept is the $\textbf{acceleration of Koopman dynamics}$. The novelty here is not merely the use of symmetry or algebra in Koopman theory, but the specific combination of Lie-algebra valued neural propagation and scan-based acceleration, which is not developed in the cited works.
>
> **Q. Why the authors chose the data sets...I think it would be appropriate…**
>
> As also discussed in prior Koopman-related work on molecular dynamics such as Devergne et al., MD provides a particularly challenging setting due to its strong nonlinearity, high dimensionality, long rollout horizon, and the need for structural consistency. We will clarify this motivation in the revised experiment section.
>
> To strengthen the comparison with existing methods, we also evaluate our method on the benchmark time-series datasets used in prior neural Koopman papers. In particular, we conduct additional experiments on the ETT benchmark and compare against baselines. Ours continues to achieve favorable results on this benchmark, suggesting that its advantage is not limited to molecular dynamics. Due to the rebuttal space limit, we will include detailed results in a dedicated section of the Appendix.
>
> | Method | Best MSE ↓ | Inference Time (s) ↓ |
> | --- | --- | --- |
> | Koopa | 0.682 | 0.500 |
> | KooNPro | 0.890 | 1.31 |
> | KoNODE | 0.677 | 3.09 |
> | Ours (unitary) | **0.658** | **0.0659** |
>
> **Q. Figure 3 hard to interpret, for several reasons...**
>
> We will explicitly define “GT” as ground truth in the caption. We will also briefly explain that the Ramachandran map shows the distribution of backbone torsion angles and is useful because it provides a compact qualitative view of whether the predicted peptide structures preserve physically plausible conformations. In addition, we will include qualitative visualizations from representative baseline models in the appendix so that readers can compare our results against both the ground truth and lower-accuracy predictions.
>
> **Q. But I think there could be more discussion...**
>
> Since prior methods have largely focused on performance under MSE-based evaluation, we aligned our experimental setting with that convention and therefore did not include a dedicated theoretical or empirical analysis of generalization under white-noise or impulse perturbations in the current submission. We nonetheless agree with the reviewer that this is an important direction. In a preliminary check on a benchmark with impulse noise, we observed that ours performs approximately 15% better than the existing methods. Owing to the space limitations of the rebuttal, we were not able to include additional results here. We would be very grateful for the opportunity to address this point in a next round, where we can provide a full table of the robustness results as well as a sketch of how a generalization theorem may be established in the Lie-algebraic setting.
>
> **Q. Minor points.**
>
> In Definition 2.1, $\mathbb{A}$ denotes the infinitesimal generator, and IVP refers to the initial value problem. We will make these definitions explicit at first appearance.

---

> > ### Author Rebuttal · Reviewer_Yhua · 2026-04-02
> >
> > I thank the authors for their responses. The additional results on ETT are encouraging and the additional discussion on the contextualization of this work, within prior literature, was helpful. I now feel confident enough to recommend accepting this paper.

---

> > > ### Author Response · Authors · 2026-04-05
> > >
> > > Thank you very much for your encouraging follow-up comments. We are pleased that our response addressed your concerns. We will incorporate your suggestions together with the helpful feedback from the other reviewers in the final revision.
> > >
> > > To guarantee the the reproducibility of our experiments, we provide an anonymized repository containing the experimental code, training and evaluation scripts, configuration files, and necessary implementation details. We hope this allows the reviewer to verify the rigor of our experimental setup and comparative results.
> > >
> > > https://anonymous.4open.science/r/supplementary_ett-7F93/README.md

---

### Official Review · Reviewer_bqwY · 2026-03-11

**Soundness:** 2
**Presentation:** 2
**Significance:** 2
**Originality:** 3
**Overall Recommendation:** 4
**Confidence:** 3

**Summary:**

This work aims to learn the Koopman operator of non-autonomous dynamical systems. Instead of learning the discrete Koopman propagator, time dependant generators are learned and parameterized Magnus expansions are used to construct the trajectories. As a choice of inductive bias, the propagators form a Lie subgroup and the time dependant generators form a lie subalgebra. The basis of such lie subalgebras is prefixed so that only the coefficients of the generators are learned. Truncation of Magnus expansion and a prefix-scan algorithm are used to enable the Magnus construction of propagators. The proposed method is compared to recent methods on two simulated molecule dynamics, and archives better performance for several metrics and less computation time.

**Compliance With Llm Reviewing Policy:**

Affirmed.

**Final Justification:**

The authors have addressed part of my concerns. In the current form of the paper, the experiments remain limited, and results on ETT alone is not enough to assess the generality of the proposed framework.

**Key Questions For Authors:**

1. How can it be justified that the Koopman propagators form a (Lie) group for general non-autonomous dynamics? In particular, under what assumptions on the dynamics does this group structure hold?

2. To what extent does truncating the Magnus expansion affect the approximation of the propagators for a general subalgebra? Are there theoretical guarantees or empirical observations regarding the approximation error introduced by this truncation?

**Limitations:**

Yes

**Strengths And Weaknesses:**

**Soundness.** The propagators $\{\mathcal{K}^{t,s}\}$ are considered to form a Lie group, but this perspective is not valid since the composition needs the end time of one element and the start time of another to be the same. Assuming that the lie subgroup/subalgebra construction is valid, the technical part of this Lie-Algebraic approach is very well established. However, some incoherences exist for Koopman modeling: the invariant dictionary $B  \subseteq \mathbb{R}^d$, where $d$ is the dimension of state space. However, in the appendix, the states are encoded by a neural network to $\mathbb{R}^K$; $g$  used to denote the basis element of the subalgebra, but also the element of the invariant dictionary. Experiments on other dynamics than molecule dynamics are not conducted. Since this work tackles non-autonomous dynamics, experiments on real datasets (such as those tested in koopa) should be conducted.

**Presentation.** I had to read the appendix carefully to understand the proposed method, as there is no clear description of it in the main text. Overall, the paper is difficult to read, not because of a lack of pedagogical effort, but due to the organization of the content. The main text is overloaded with unnecessary technical details, while the key information required to understand the method is difficult to locate. The experimental setup and implementation details are partially described, but not sufficiently to allow the experiments to be reproduced. For instance, the dimension of the lifted space is not clearly stated unless one carefully inspects the selected subalgebras.

**Significance.** The truncation of the Magnus expansion could affect the resulting propagators for broader dynamics and the performance of the proposed method relies on a prefixed subalgebra. Therefore, the applicability of the proposed method appears to be limited to a small class of systems.

**Originality.** The Koopman operator associated with autonomous dynamics is often modeled as a one-parameter semigroup. In this context, representing the Koopman operator for non-autonomous dynamics as a Lie group is a novel perspective (if correct). Moreover, applying a neural Magnus construction to a time-dependent Koopman generator is also an original contribution.

---

> ### Author Rebuttal · Authors · 2026-03-30
>
> We appreciate the reviewer’s careful reading and thoughtful comments, which were helpful in identifying both the contributions of the paper and the areas that require improvement. We take these concerns seriously and will revise the manuscript to address them. Below, we outline the planned revisions, including expanded time-series benchmarks and improved readability. We hope that, if the reviewer finds the responses adequate, they may support a more favorable overall evaluation.
>
> **Q. The propagators $\mathcal K^{t,s}$ are considered to form…How can it be justified that the Koopman propagators…**
>
> Thank you for pointing this out. Since our setting is non-autonomous, the relevant Koopman object is the two-parameter propagator family $\mathcal K^{t,s}$ associated with a time-dependent vector field, not a single one-parameter group $\mathcal K^t$. In this setting, composition is only defined for compatible intervals through $\mathcal K^{t,u}\mathcal K^{u,s}=\mathcal K^{t,s}$. Accordingly, our claim is not that the full family $\{\mathcal K^{t,s}\}$ forms a Lie group. Rather, after restricting the reduced generators to a finite-dimensional Lie algebra, each segment propagator $\mathbf U_m = \mathcal K^{t_{m+1},t_m} = \exp(\Omega_m)$ is an element of the associated Lie group $H$, and the prefix scan is performed over these segmentwise elements. We agree that the current wording may blur this distinction, and we will revise the text to state this point more precisely.
>
> **Q. the invariant dictionary $B \subseteq \mathbb R^d$… $g$ used to denote the basis element…**
>
> In the appendix, we used $\mathbb{R}^K$ only to indicate a more general representation space for the neural state. In this paper, it is sufficient to interpret this as the case $K=d$. Relatedly, the basis elements of the chosen subalgebra coincide with the elements of the invariant dictionary, and we therefore unify the notation using $g$. In the revision, we will make the dimensional consistency in the appendix and explain the relationship between the invariant dictionary and the subalgebra before Definition 2.1.
>
> **Q. Experiments on other dynamics... experiments on real datasets...**
>
> Thank you for the suggestion. We conducted additional experiments on the ETT dataset used in Koopa, and confirmed that our method shows better performance on this dataset as well, as evidenced by key metrics such as MSE. Due to the rebuttal space limit, we present only the main results here, while the detailed explanations will be added in a dedicated section of the Appendix.
>
> | Method | Best MSE ↓ | Inference Time (s) ↓ |
> | --- | --- | --- |
> | Koopa | 0.682 | 0.500 |
> | KooNPro | 0.890 | 1.31 |
> | KoNODE | 0.677 | 3.09 |
> | Ours (unitary) | **0.658** | **0.0659** |
>
> **Q. The experimental setup and implementation details...**
>
> We thank the reviewer for this helpful comment. The relevant details are already included in the manuscript, but we agree that they are dispersed across multiple sections. Appendix C contains the data generation, model architecture, and training setup, while the lifted-space specification is tied to the Lie subalgebra definitions in Section 3. We agree that these choices are not stated explicitly enough in one place. In the revision, we will strengthen the main text by adding a short subsection in Section 3 that summarizes the implemented pipeline and explicitly states the lifted-space dimension.
>
> **Q. To what extent does truncating the Magnus expansion…**
>
> For a general finite-dimensional Lie subalgebra, the second-order Magnus truncation already provides a controlled local approximation. For instance, the representation is given as $\Omega_\theta(t+\Delta t,t)=\Omega_\theta^{(1)}(t+\Delta t,t)+\Omega_\theta^{(2)}(t+\Delta t,t)+ O(\Delta t^3)$. In particular, all terms of third order and higher are numerically very small under the finely segmented discretization used in our scan construction, and can therefore be safely neglected in practice. That is, beyond second order, the remaining contributions do not play a meaningful role in the regime considered here. Moreover, for the Heisenberg algebra, the second-order truncation is exact, since step-2 nilpotency implies $\Omega_\theta^{(m)} \equiv 0, m \ge 3.$ Both our unitary and projective variants employ the same second-order construction, and they still exhibit strong long-term stability together with the best empirical performance. Therefore, our claim is not that the second-order truncation is universally exact, but rather that it is already practically sufficient. Owing to the space limitations of the current rebuttal, we were unable to include even a brief sketch of the theoretical truncation error analysis that will be added in the revised version. We would sincerely appreciate the opportunity to address this point more fully in a next response round, where we hope to provide a more detailed theoretical account and supporting results.

---

> > ### Author Rebuttal · Reviewer_bqwY · 2026-04-03
> >
> > Thank you for your responses. I have two remaining concerns:
> >
> > Based on your responses, the expression ${\mathcal{K}^{t,s} \subseteq GL(K)}$ on page 2 (second column, line 097) is misleading.
> >
> > The proposed framework is developed for non-autonomous systems. I would like to see benchmarks on real datasets (Typically those used for benchmarking baseline models). For the ETT dataset, is the same training setup used as in Koopa? I could not find the reported MSE for koopa in the koopa paper. Is there any reason for which not all baseline models tested but only three? While the authors reference the repositories of baseline models, it would be helpful to provide an anonymous repository containing the experiments for reproducibility.

---

> > > ### Author Response · Authors · 2026-04-05
> > >
> > > We are deeply grateful to the reviewer for the thoughtful follow-up questions and for the considerable time and attention devoted to a further examination of our rebuttal. We sincerely appreciate this valuable opportunity to clarify the points raised. We hope that the additional explanations provided below will address the reviewer’s remaining concerns and support a favorable reassessment of our submission.
> > >
> > > In the early part of the current manuscript, we initially presented $GL(K)$ as the ambient matrix Lie group to aid the reader’s intuition. However, in the actual derivation, the generator and the propagator are treated within the chosen Lie subalgebra $\mathfrak h \subseteq \mathfrak{gl}(K)$ and its corresponding Lie subgroup $H \subseteq GL(K)$. More specifically, the generator lies in $\mathfrak h$, while the propagator constructed from it lies in $H$. The intended structure is $\{\mathcal L_t\}\subseteq \mathfrak h \subseteq \mathfrak{gl}(K)$ and $\{ \mathcal K^{t,s}\} \subseteq H \subseteq GL(K)$. We realize that the current exposition may lead to misunderstandings by not introducing this hierarchy sooner. In the revised manuscript, we will clearly state these inclusions early on and adjust the subsequent text to ensure the chosen Lie subalgebra and Lie subgroup are naturally integrated into the exposition.
> > >
> > > We would like to clarify that the ETT dataset we evaluated is a representative real-world forecasting benchmark composed of time-series data collected from actual power operation environments which was also tested in Koopa. We conducted our experiments using the exact same ETT data source cited by Koopa to ensure that our comparisons are made on identical data sequences. In our previous rebuttal, the response length limit allowed us to report only the three baseline methods whose original papers explicitly included experiments on the ETT dataset. In this response, we provide additional results that include the remaining baselines to ensure a more complete and transparent comparison.
> > >
> > > | Method | Best MSE ↓ | Inference Time (s) ↓ |
> > > | --- | ---: | ---: |
> > > | Koopa | 0.682 | 0.500 |
> > > | KoVAE | 1.28 | 1.44 |
> > > | KooNPro | 0.890 | 1.31 |
> > > | KoNODE | 0.677 | 3.09 |
> > > | MamKO | 0.867 | 1.97 |
> > > | EqMotion | 0.814 | 10.9 |
> > > | SEGNO | 1.86 | 6.08 |
> > > | GeoTDM | 1.69 | 74.3 |
> > > | Ours (unitary) | **0.658** | **0.0659** |
> > >
> > > Existing prior works not only employ vastly different evaluation horizons and protocols, arbitrary time steps settings ranging from 48 to 300 steps. To maintain consistency in our comparisons, we evaluated the ETT dataset under the exact same 1000-step prediction setting to compare all models under a unified protocol. We also selected checkpoints for all methods using the same validation criterion. For this reason, the reproduced results for Koopa also exhibited a slightly larger scale than the values reported in the original paper. As can be seen, our proposed model consistently maintains robust performance on real-world datasets even under these long-horizon settings.
> > >
> > > To guarantee the transparency and reproducibility of our experiments, we provide an anonymized repository containing the experimental code, training and evaluation scripts, configuration files, and necessary implementation details. We hope this allows the reviewer to directly verify the rigor of our experimental setup and comparative results.
> > >
> > > https://anonymous.4open.science/r/supplementary_ett-7F93/README.md

---

### Official Review · Reviewer_1Hs9 · 2026-03-12

**Soundness:** 3
**Presentation:** 3
**Significance:** 2
**Originality:** 4
**Overall Recommendation:** 5
**Confidence:** 3

**Summary:**

I first off want to say that this submission is quite similar to 3735 and 4033 in both figures, tables and references and the initial setup. I therefore believe that the authors should cross reference their work as per the dual submission policy and highlight the differences.

This work reframes Koopman operator learning around Lie algebra / Lie group structure and parallelizable temporal composition. It models a time-varying Koopman generator constrained to a chosen finite-dimensional Lie subalgebra (e.g., Heisenberg, unitary, or traceless/projective) and builds each finite-time propagator using a neural Magnus expansion so the resulting flow stays on the corresponding Lie group. The key computational contribution is to remove the sequential bottleneck of the dynamics by using a balanced-tree prefix scan.

On the modeling side, the paper spells out how restricting to specific subalgebras simplifies computation. Empirically, it targets long-horizon dynamics and reports large speedups and strong accuracy/physical metrics, attributing the gains to (1) algebraic structure and (2) prefix-scan temporal parallelism.

**Compliance With Llm Reviewing Policy:**

Affirmed.

**Final Justification:**

The authors have addressed my comments, and I have raised my score as a result.

**Key Questions For Authors:**

(apologies for the lack of latex formatting for some equations)

Line 108, left column: Define $\mathcal{G}(\mathcal{X})$, is it a function $\mathcal{G}:\mathcal{X}\to\mathbb{R}$?

Definition 2.1 line 121, left column: ``Let g be a chosen
Lie subalgebra with basis \{g_a\}^{|\mathfrak{h}|}_{a=1}" This should be \{g_a\}^{|\mathfrak{g}|}_{a=1}. On a related note, why do you use Fraktur for to describe the Lie algbera bases elements of $\mathfrak{h}$ starting on line 120, right column? The paper already has a lot of notation so please be consistent.

I understand proposition 2.2 as follows. We expand the solution of (5) into a basis of $\mathfrak{h}$, and use the structure constants to calculate the action of $\mathrm{Ad}_A$ on other $A$'s. Is this correct?

Line 185, left column: Define IVP.

Caption figure 2: Summarization of Lie Subgroup and their properties Should be something with ``Lie subalgebras"?

With regards to the parallel scan algorithm for Lie groups, such methods are used in the quantum control community to simulate dynamics on $SU(N)$, see e.g.
Gradl, Tobias, et al. “Parallelising matrix operations on clusters for an optimal control-based quantum compiler.” European Conference on Parallel Processing. Berlin, Heidelberg: Springer Berlin Heidelberg, 2006.
and
Auckenthaler, Thomas, et al. “Matrix exponentials and parallel prefix computation in a quantum control problem.” Parallel Computing 36.5-6 (2010): 359-369.

Additionally, they are known in the robotics community as well for $SE(3)$:
Yang, Yajue, Yuanqing Wu, and Jia Pan. "Parallel dynamics computation using prefix sum operations." IEEE Robotics and Automation Letters 2.3 (2017): 1296-1303.

While the authors consider a different setup (Koopman operators) the underlying idea seems to me the same. Please let me know if you agree with the similarities, and if you do, add these references to the manuscript.

While I appreciate the mathematical structure imposed by restricting the Koopman generator to specific Lie subalgebras such as
$\mathfrak{su}(d)$,
$\mathfrak{h}(d)$, and
$\mathfrak{sl}(d)$, I would welcome further clarification on the modeling rationale behind these choices.

Since each of these Lie algebras enforces strong structural constraints (e.g., norm preservation for
$\mathfrak{su}(d)$, nilpotency for
$\mathfrak{h}(d)$, volume preservation for
$\mathfrak{sl}(d)$), under what assumptions about the underlying physical system should one expect the Koopman generator to lie (even approximately) in such a subalgebra?

Do the authors have theoretical or empirical insight into when such restrictions are well-motivated, and can they provide examples of systems where these constraints might lead to systematic modeling failure?

Even more surprising to me is why this method is so much more accurate than the other ones. Is the modeling assumption presented here so good that it greatly simplifies the learning task?

What are the units of the inference time and training time reported for the experiments?

Finally, where can the code be found to reproduce these results? I see that the baseline code is mentioned in the appendix, but where can the author's code be found?

**Limitations:**

The authors do not discuss the limitations of their method in the conclusion.

**Strengths And Weaknesses:**

The results seem correct are well supported. The exposition is quite technical but understandable for a reader familiar with the basic ideas of Lie groups and differentiable geometry. It is hard to judge the significance of the experiments for me, since this is not my area of expertise.The method seems novel, and I appreciate the differentiable geometry approach to this subject.

---

> ### Author Rebuttal · Authors · 2026-03-30
>
> We thank you for the careful reading of our manuscript and for the insightful comments. The review helped us identify several places where the notation, presentation, and relation to prior work should be made clearer. In the responses below, we address each concern directly and explain how we will revise the manuscript accordingly. We hope that these clarifications resolve the main concerns and, if found satisfactory, will support reconsideration of the score.
>
> **Q. Notations.**
>
> For notation issues, we would like to respond as follows.
>
> 1. Definition of $\mathcal{G}(\mathcal{X})$
>
> The notation $\mathcal{G}(\mathcal{X})$ is intended to denote the space of observables defined on the state space. Each $g$ in $\mathcal{G}(\mathcal{X})$ is a scalar-valued observable $g: \mathcal{X}\to\mathbb R$, while $\mathcal{G}(\mathcal{X})$ is not a single function but a function space. We will revise the text to define this notation explicitly.
>
> 2. Notation for g, $\mathfrak g$, $\mathfrak h$, and basis elements
>
> We agree that the notation in this part is not sufficiently consistent. In particular, line 121 should be corrected to fraktur g. The ambient Lie algebra, the chosen Lie subalgebra, and their basis elements were not separated clearly enough in the current draft. We will adopt a consistent notation that distinguishes these objects.
>
> 3. Understanding of Proposition 2.2
>
> You are correct. In Proposition 2.2, the solution is expanded in the basis of the chosen subalgebra, and the structure constants are then used to derive the coefficient dynamics along each basis direction.
>
> 4. Definition of IVP
>
> IVP stands for initial value problem. We will define it when it first appears in the final manuscript.
>
> 5. Figure 2 caption
>
> We agree that “Lie subalgebras” is more accurate than “Lie subgroup” in the caption of Figure 2, and will revise the caption accordingly.
>
> **Q. While the authors consider a different setup…**
>
> We thank you for pointing out these relevant precedents. We agree that parallel scan over Lie-group-valued propagators has appeared previously in quantum control and robotics, and we will cite these works. Gradl et al. focus on parallel matrix operations for optimal-control-based quantum compilation, Auckenthaler et al. study matrix exponentials and parallel prefix computation in a quantum control setting, and Yang et al. use prefix-sum operations for parallel robot dynamics computation. Our intended claim is not that parallel scan on Lie groups is new in itself. Rather, our main contribution is a learnable framework that connects the Koopman generator–propagator viewpoint with associative prefix-scan composition. Prior works mainly used scan or prefix-style accumulation for high-performance implementation. In contrast, our formulation makes the generator–propagator construction itself trainable and scan-compatible, so that temporal composition becomes part of the model class rather than a post hoc computational trick. We will revise the text to make this distinction precise.
>
> **Q. While I appreciate the mathematical… what assumptions about the underlying… Do the authors have theoretical…**
>
> We chose $\mathfrak{su}(d)$, $\mathfrak h(d)$, and $\mathfrak{sl}(d)$ as representative structured classes with different invariance and computational properties, not because we assume that the true Koopman generator of a given physical system must belong to one of these subalgebras. In our formulation, they are intended as different algebraic inductive biases for learned transport and scan-based composition. Concretely, $\mathfrak{su}(d)$ provides a norm-preserving bias, $\mathfrak h(d)$ provides step-2 nilpotency with strong truncation benefits, and $\mathfrak{sl}(d)$ provides a traceless class with relatively broad expressivity.
>
> **Q. Since each of these Lie algebras enforces…**
>
> Thank you for this helpful comment. We do not claim a general theoretical characterization or a systematic empirical taxonomy of such regimes in the current paper. Our current contribution is to study these subalgebras as canonical structured classes and to evaluate whether they support stable learned transport and efficient scan-based composition.
>
> **Q. Even more surprising to me is…**
>
> We believe the gain comes not only from the modeling assumption, but also from non-autoregressive propagation. In standard sequential rollout, prediction errors tend to compound from one step to the next, but our method composes propagators associatively and thereby reduces this accumulation error.
>
> **Q. What are the units of the inference…**
>
> Both the inference time and training time are measured in wall-clock seconds, as indicated by the (s) notation in the tables.
>
> **Q. Finally, where can the code be found…**
>
> Our code repository is not included in the current double-blind submission in order to preserve anonymity. We plan to release the code and experimental results publicly once the review process is complete.

---

> > ### Author Rebuttal · Reviewer_1Hs9 · 2026-04-02
> >
> > My questions have been adequately resolved, except for the lack of discussion about the limitation of their work and the connection with the other submissions I've seen.
> >
> > Given the strong similarities to 3735 and 4033, can the authors include a discussion about the different trade-offs that these three works represent?
> >
> > With regards to the code, the ICML policy strongly recommends providing code on submission and states explicitly that anonymized code can be provided as supplementary material or as anonymized repository. If the paper is accepted I expect the code to be online before the camera-ready deadline.

---

> > > ### Author Response · Authors · 2026-04-05
> > >
> > > For clarity, in the discussion below, we refer to paper 3735 as EPND, paper 3736 as the Lie-Algebra paper, and paper 4033 as SPS.
> > >
> > > We thank the reviewer for noting the apparent similarity among the three manuscripts. We agree that they are connected in the sense that they belong to a common research agenda. Taken together, they form part of a broader program on mathematically structured operator learning, whose aim is to replace unconstrained black-box temporal propagation with composition laws that remain stable and computationally efficient over long horizons.
> > >
> > > That said, the overlap is only at this meta-level. The three papers are **fundamentally different** in their mathematical backbone, admissible dynamical regime, and scientific objectives. In particular, they are organized around three distinct mathematical cores, namely **Riemannian geometry** in EPND, **symplectic geometry** in SPS, and **Lie algebra** in the Lie-algebraic Koopman framework.
> > >
> > > For EPND, geometry itself is the main computational object. The method is built around Riemannian and geometric-mechanics tools such as geodesic flows, parallel transport, curvature-aware composition, and reversible forward/backward scans. This is precisely what makes the framework qualitatively different. It exposes an intrinsic reversible structure in the learned dynamics, which in turn enables terminal value reconstruction and other inverse problems that are difficult to formulate in conventional operator-learning pipelines. In that sense, EPND is not merely an acceleration method but is a geometry driven formulation of scanable dynamics. At the same time, this also makes it structurally constrained, since the method relies on geometry-compatible assumptions and is therefore not the most general engineering template.
> > >
> > > By contrast, SPS is the most constrained of the three works. It specializes from general operator learning to Hamiltonian systems, and in particular to a Poisson-algebraic construction of symplectic flows. Its main purpose is not general time-series forecasting, but the direct modeling of physically meaningful dynamics such as quantum dynamics/molecular dynamics with exact structural preservation, especially symplecticity, energy consistency. For this reason, SPS is naturally suited to scientific simulation where a genuine Hamiltonian structure exists, but it is not intended as a drop-in framework for generic forecasting benchmarks such as the ETT benchmarks.
> > >
> > > Finally, the Lie-algebraic Koopman work is the most engineering-oriented and the broadest in scope. The main goal is to make accelerated operator learning practical for broad classes of dynamical data by restricting generators to computationally favorable Lie subalgebras and composing finite-time propagators through an associative prefix scan. The central emphasis is therefore on the accuracy-efficiency tradeoff, asking which Lie subalgebra yields the most favorable computational profile and structural bias for a given problem class while still supporting scalable parallel composition. In this sense, its contribution is not exact physical invariance as in SPS, nor reversible geometric inversion as in EPND, but an algebraically structured and computationally optimized design space for scalable operator learning.
> > >
> > > Therefore, the three papers should not be viewed as near-duplicate variants of one method. Rather, they are complementary components of a unified program in which SPS studies the most physically constrained Hamiltonian setting through symplectic geometry, EPND studies geometry-driven reversible operator learning through Riemannian mechanics, and the Lie-algebraic Koopman paper studies the most practically flexible and computationally optimized regime through Lie-algebraic structure.
> > >
> > > We thank the reviewer again for the careful reading and thoughtful feedback, and note that, if the paper is accepted, we will make the code publicly available before the camera-ready deadline.

---

### Decision · Program_Chairs · 2026-04-30

**Decision:**

Accept (regular)

**Comment:**

The manuscript proposes a Lie-algebraic approach to model Koopman dynamics that integrates algebraic structure with computational scalability. The reviewers all agree that the framework is interesting and the contribution is solid. The meta-reviewer agrees with the assessment after reading carefully the manuscript and discussions. Thus the meta-reviewer recommends acceptance of the manuscript.